# A cytoplasmic peptidoglycan amidase homologue controls mycobacterial cell wall synthesis

Cara C Boutte[1], Christina E Baer[2], Kadamba Papavinasasundaram[2], Weiru Liu[1], Michael R Chase[1], Xavier Meniche[2], Sarah M Fortune[1], Christopher M Sassetti[2], Thomas R Ioerger[3], Eric J Rubin[1,4]*

[1]Department of Immunology and Infectious Diseases, Harvard TH Chan School of Public Health, Boston, United States; [2]Department of Microbiology and Physiological Systems, University of Massachusetts Medical School, Worcester, United States; [3]Department of Computer Science, Texas A and M University, Texas, United States; [4]Department of Microbiology and Immunobiology, Harvard Medical School, Boston, United States

**Abstract** Regulation of cell wall assembly is essential for bacterial survival and contributes to pathogenesis and antibiotic tolerance in *Mycobacterium tuberculosis (Mtb)*. However, little is known about how the cell wall is regulated in stress. We found that CwlM, a protein homologous to peptidoglycan amidases, coordinates peptidoglycan synthesis with nutrient availability. Surprisingly, CwlM is sequestered from peptidoglycan (PG) by localization in the cytoplasm, and its enzymatic function is not essential. Rather, CwlM is phosphorylated and associates with MurA, the first enzyme in PG precursor synthesis. Phosphorylated CwlM activates MurA ~30 fold. CwlM is dephosphorylated in starvation, resulting in lower MurA activity, decreased cell wall metabolism, and increased tolerance to multiple antibiotics. A phylogenetic analysis of *cwlM* implies that localization in the cytoplasm drove the evolution of this factor. We describe a system that controls cell wall metabolism in response to starvation, and show that this regulation contributes to antibiotic tolerance.

*For correspondence: erubin@hsph.harvard.edu

**Competing interests:** The authors declare that no competing interests exist.

## Introduction

*Mycobacterium tuberculosis (Mtb)*, a bacterium that causes 1.5 million deaths a year (*Sharma et al., 2006*; *World Health Organization, 2014*), differs from many bacterial pathogens in its infection strategy, which depends on an unusual cell envelope (*Cambier et al., 2014*). The importance of this structure is evident from the fact that at least a quarter of the genes in the *Mtb* genome are involved in its construction and regulation (*Doerks et al., 2012*). The mycobacterial cell wall consists of peptidoglycan (PG) covalently bound to a layer of arabinogalactan sugars, which is in turn covalently bound to a layer of mycolic acid lipids that make up an outer membrane (*Alderwick et al., 2015*; *Cambier et al., 2014*).

While the chemistry of this cell envelope is increasingly well defined, its regulation is poorly understood. Regulated cell wall changes are essential for *Mtb* to cause disease (*Doerks et al., 2012*; *Sharma et al., 2006*) and are thought to contribute to phenotypic antibiotic tolerance. TB therapy requires 6–12 months of drug treatment. This is likely due, in part, to drug tolerance like that observed in stressed, non-growing Mtb cultures (*Wallis et al., 1999*). Because stressed Mtb also exhibits cell wall changes such as thickening, differential staining and chemical alterations (*Bhamidi et al., 2012*; *Cunningham and Spreadbury, 1998*; *Seiler et al., 2003*), it seems probable

**eLife digest** Bacterial cells are surrounded by a protective cell wall. Some bacteria, including the species *Mycobacterium tuberculosis* that causes tuberculosis, have the ability to change the properties of their cell wall when exposed to stressful conditions. These changes may also help the bacteria to resist the antibiotics used to treat the infections they cause. However, little is known about how changes to the cell wall are regulated.

By carrying out genetic experiments in a non-infectious relative of *M. tuberculosis* and by performing biochemical assays with *M. tuberculosis* proteins, Boutte et al. have now investigated the role of a bacterial protein called CwlM. This protein was predicted to be an enzyme that cuts peptidoglycan, a network of sugars and short proteins that forms part of the bacterial cell wall. Such an enzyme would allow the peptidoglycan to expand and remodel. However, Boutte et al. found that CwlM does not act as an enzyme. Instead, it regulates an enzyme called MurA, which is present inside bacteria and helps to make the peptidoglycan network. Thus, CwlM actually helps to determine how much peptidoglycan a cell produces.

When activated, the CwlM protein binds to MurA and stimulates it to start producing peptidoglycans. However, Boutte et al. observed that CwlM is active only when cells have plenty of nutrients. When nutrients are scarce, CwlM deactivates and reduces the activity of the MurA enzyme. This quickly shuts down the production of peptidoglycan and makes starved cells tolerant to antibiotics. Notably, increasing peptidoglycan production in starved cells, by enhancing the activity of the MurA enzyme, makes the cells more vulnerable to several antibiotics.

Future work could now investigate the conditions under which CwlM is activated and deactivated in *M. tuberculosis* during an infection. It also remains to be seen whether other enzymes are regulated in a similar way to MurA.

that the cell wall changes contribute to antibiotic tolerance. Importantly, starved, antibiotic tolerant Mtb cells have been shown to be less permeable to antibiotics (*Sarathy et al., 2013*). However, the regulatory mechanisms that induce cell wall changes in response to stress and contribute to impermeability and tolerance have not been described.

The peptidoglycan (PG) layer provides protection and shape-defining structure to cells of almost all bacterial species. This layer is constructed by transpeptidases and glycosyltransferases, which attach new precursors to the existing cell wall; and several types of catabolic PG hydrolases, which break bonds in the existing PG. PG enzymes must be tightly regulated to promote cell growth and septation without compromising the wall integrity, and to restructure the cell wall to withstand stresses (*Kieser and Rubin, 2014*). Accordingly, a large number of regulators in the cytoplasm, inner membrane and periplasm coordinate and control the activities of the PG enzymes, either directly or indirectly (*Kieser and Rubin, 2014*; *Typas et al., 2011*).

In addition to the dedicated regulators, many PG synthases and hydrolases work together in complexes and regulate each others' enzymatic activity through protein-protein interactions (*Banzhaf et al., 2012*; *Hett et al., 2010*; *Smith and Foster, 1995*). Notably, some PG hydrolases have lost their enzymatic activity and function only as regulators: EnvC in *E. coli* is missing catalytic residues from its active site but contributes to cell septation by activating the PG amidases AmiA and AmiB (*Uehara et al., 2010*; *Yang et al., 2011*).

In this work we study the predicted PG hydrolase, CwlM, and find that its essential function is regulatory rather than enzymatic. We find that CwlM is located in the cytoplasm, is phosphorylated by the essential Serine Threonine Protein Kinase (STPK) PknB, and functions to stimulate the catalytic activity of MurA, the first enzyme in the PG precursor synthesis pathway. In nutrient-replete conditions CwlM is phosphorylated. Using *in vitro* biochemistry we show that phosphorylated CwlM (CwlM~P) increases the rate of MurA catalysis by ~30 fold. In starvation, CwlM is dephosphorylated and in this state does not activate MurA, which has very low activity alone (*Xu et al., 2014*). Importantly, we find that over-activation of MurA in the transition to starvation causes increased sensitivity to multiple classes of antibiotics. Finally, a phylogenetic analysis implies that CwlM protein evolution was driven by localization to the cytoplasm.

## Results

*cwlM* (Rv3915, MSMEG_6935) is predicted to be essential for growth in *Mtb* (*Zhang et al., 2012*), is highly conserved among mycobacteria, and is annotated as an N-acetylmuramoyl-L-alanine amidase of the AmiA/LytC family (hereafter: PG amidase), a type of PG hydrolase. CwlM_{TB} has been shown to have PG amidase activity (*Deng et al., 2005*); however, key residues that coordinate the catalytic $Zn^{2+}$ are not conserved in CwlM from both *Mtb* and *Msmeg* (*Yamane et al., 2001*) (*Figure 1A*), implying that it is inefficient as an enzyme. This is supported by our observation that overexpression of CwlM did not affect cell viability (*Figure 1—figure supplement 1*): overexpression of highly active PG hydrolases usually results in cell lysis (*Uehara and Bernhardt, 2011*). Thus, we hypothesized that CwlM's essential function might not be enzymatic. Certain PG hydrolases in *E. coli* function as regulators of other PG hydrolases rather than as enzymes (*Uehara et al., 2010*). We hypothesized that CwlM may have a similar role in activating an essential enzyme.

### CwlM is essential for growth in *M. smegmatis*, but may have a non-enzymatic function

To confirm the essentiality of *cwlM* in *Msmeg* we constructed a strain (See *Supplementary file 1* for full descriptions of all strains), Ptet::*cwlM*, in which the only copy of *cwlM* is under control of an anhydrotetracyline (Atc)-inducible promoter. Depletion of CwlM by removing Atc results in cell death (*Figure 1B*). Microscopy of CwlM-depleted cells shows that they are short, implying a defect in elongation. To assess polar elongation (*Aldridge et al., 2012*; *Thanky et al., 2007*), we stained cells with an amine reactive dye (ARD) (*Aldridge et al., 2012*) and cultured them to allow new, unstained polar cell wall to form before imaging (*Figure 1C*). We found that CwlM-depleted cells fail to elongate normally (*Figure 1D,E,F*).

To determine if CwlM requires amidase activity for its essential function, we mutated the active site aspartate and glutamate that coordinate the catalytic $Zn^{2+}$ (*Prigozhin et al., 2013*) and have been shown to be required for catalysis in related proteins (*Prigozhin et al., 2013*; *Shida, 2001*). We performed allele swapping at the L5 phage integrase site (*Pashley and Parish, 2003*) to replace the wild-type (WT) *cwlM* with the mutant allele *cwlM* E209A D331A. *cwlM* E209A D331A has zero out of four conserved $Zn^{2+}$-coordinating residues, but is able to complement the WT allele in a growth curve assay (*Figure 1G*). Thus, CwlM is essential for cell survival and elongation in *Msmeg*, but its presumed amidase active site does not appear to be essential. CwlM may therefore have another function.

### CwlM is phosphorylated, and this phosphorylation is important for cell growth

A proteomic screen in *Mtb* found that CwlM_{TB} is phosphorylated at T43 and T382 (*Prisic et al., 2010*). To determine whether the *Msmeg* CwlM is also phosphorylated, we constructed a strain with an epitope-tagged allele (*cwlM*::FLAG), immunoprecipitated the protein from *Msmeg* lysate and separated it with 2D gel electrophoresis. We found several CwlM species with different isoelectric points, a hallmark of phosphorylation (*Figure 2A*). Mass spectrometry of the gel spots identified phosphorylation at T35 and T374 (equivalent to T43 and T382 in the *Mtb* protein), as well as T376 and T378, and acetylation at K362 and K369. Most of the modifications are found at the C-terminal tail of the protein, which is highly conserved across actinomycetes (*Figure 2B*), despite being outside of a conserved domain. To assess whether phosphorylation is important for function, we replaced WT *cwlM* at the L5 site with mutants in which phosphorylated threonine residues were replaced by alanines. Most mutants grew normally but the L5::*cwlM* T374A strain grew slowly (*Figure 2C*; *Figure 2—figure supplement 1A*), despite containing normal amounts of CwlM protein (*Figure 2D*). The L5::*cwlM* T374A cells were short (*Figure 2E*) and defective for polar elongation (*Figure 2F*). Because the phenotypes of the L5::*cwlM* T374A strain are similar to the CwlM depletion (*Figure 1*), it seems likely that absence of phosphorylation on T374 inhibits the essential function of CwlM, though the T374A mutation could interrupt CwlM function in another way.

Peptidoglycan hydrolases and their direct regulators are found in the periplasm; however, CwlM lacks classical sec or tat secretion signal sequences (*Bagos et al., 2010*; *Petersen et al., 2011*). Moreover, phosphorylation of periplasmic proteins has not been described. To determine the compartmental location of CwlM, we used the Substituted Cysteine Accessibility Method (SCAM)

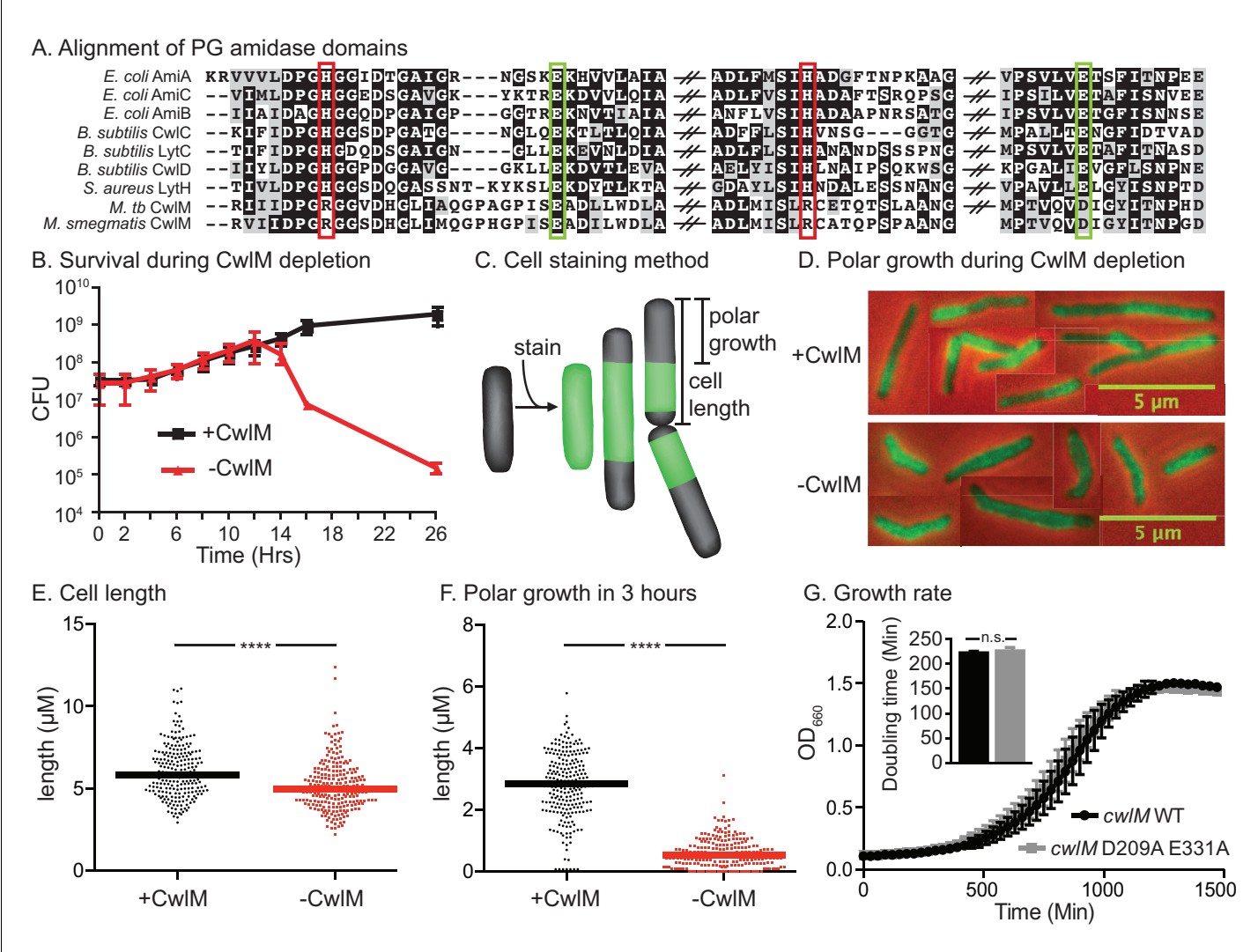

**Figure 1.** CwlM's essential function promotes polar growth, but not through enzymatic activity. (**A**) Alignment of the active-site proximal regions of the enzymatic domain of PG amidases. The $Zn^{2+}$-coordinating residues are boxed: the conserved aspartate/ glutamates in green, and the degenerate histidine/ arginines in red. (**B**) Colony forming units (CFU) of the Ptet::*cwlM* (CB82) strain grown with (+CwlM) or without (−CwlM) Atc inducer from Time 0. Data from all experiments throughout is from three biological replicates and error bars are the standard deviation, unless otherwise noted. (**C**) Cartoon of the amine reactive dye (ARD) staining method showing polar growth after staining, with a diagram of how cell length and polar growth were measured for Figures 1EF and 2EF. (**D**) Overlaid micrographs of the Ptet::*cwlM* strain grown with (+CwlM) or without (−CwlM) Atc for 9 hr, stained with ARD, grown 3 hr and imaged with a 488/530 filter (green) and in phase (red and black). Representative cells from several images used in **E** and **F** were pasted together, the scale is conserved between images. (**E**) The length of ~230 cells from each condition in (**D**). Two biological replicates were used for all microscopy experiments, unless noted. (**F**) The length of the longer unstained pole of the cells in (**E**), line indicates the median in (**E**) and (**F**). **** p value <0.0001. (**G**) Growth curves showing the $OD_{660}$ of WT (L5::*cwlM*, CB236) and amidase-ablated (L5::*cwlM* D209A E331A, CB239) strains growing in 7H9, and a bar graph showing the calculated doubling times of these strains (inset). n.s. – not significant. The p value was 0.51 by the student's t-test

The following figure supplement is available for figure 1:

**Figure supplement 1.** Overexpression of CwlM.

(*Karlin and Akabas, 1998*). We measured the accessibility of a cysteine on CwlM to agents with different membrane permeability and found that CwlM is only accessible to agents that can reach the cytoplasm (*Figure 2G*). Thus, CwlM is a cytoplasmic protein whose essential function is likely activated by phosphorylation.

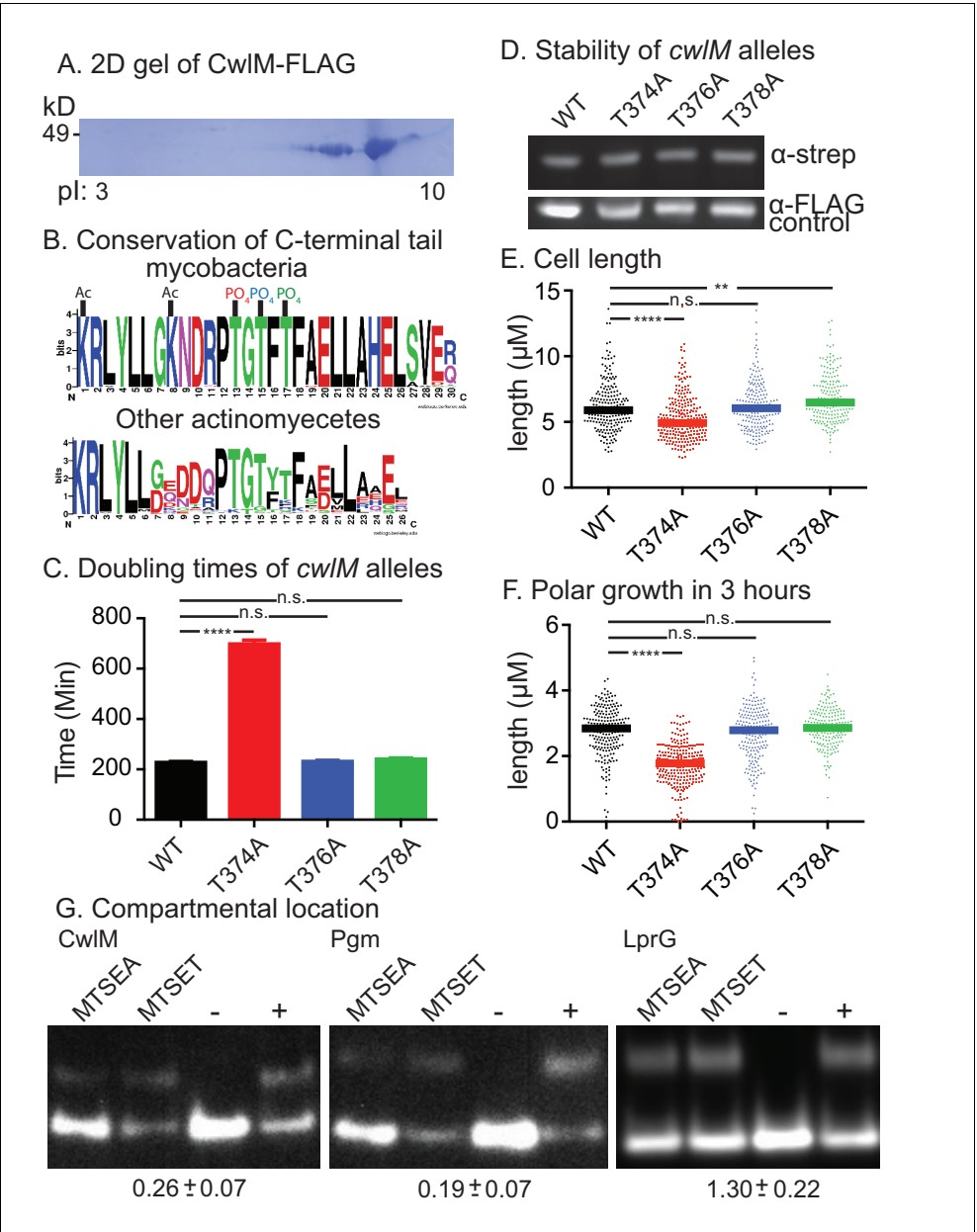

**Figure 2.** CwlM is phosphorylated and cytoplasmic. (**A**) 2-D gel of CwlM-FLAG from *Msmeg* (CB100) with the isoelectric point (pI) indicated on the bottom. (**B**) Weblogo diagrams of the C-terminus of CwlM homologues. Acetylation (Ac) and phosphorylation (PO₄) sites are indicated. The colors of the three phosphates match the colors of the corresponding phospho-ablative strains in **C**, **E** and **F**. (**C**) Doubling times of L5::*cwlM*-FLAG WT (CB300) and phospho-ablative mutants (T374A = CB300; T376A = CB345; T378A = CB348). All multiple comparisons throughout were performed by One-way ANOVA with a Dunnett multiplicity correction. The adjusted p values for each mutant compared to WT are: T374A =< 0.0001; T376A = 0.88; T378A = 0.15. (**D**) α-strep western blot of L5::*cwlM*-strep WT (CB663) and phospho-ablative mutants (CB666, 669, 672). A background band on the same blot probed with α-FLAG is used as a control. Westerns of the L5::*cwlM*-FLAG strains could not be used because of high background. (**E**) Cell length of 200–300 cells from L5::*cwlM*-FLAG WT and phospho-ablative mutants. The adjusted p values for each mutant compared to WT are: T374A =< 0.0001; T376A = 0.99; T378A = 0.01. (**F**) Length of the longer unstained pole of cells in (**E**). Cells were stained with ARD and grown for three hours before imaging. The adjusted p values for each mutant compared to WT are: T374A =< 0.0001; T376A = 0.59; T378A = 0.36. (**G**) Substituted cysteine accessibility. α-strep western of cells with strep-tagged Pgm (CB886, MSMEG_4579, cytoplasmic control), lprG (CB706, Rv1411, periplasmic control) and CwlM1cys (CB457) that were cysteine-blocked with MTSEA (membrane permeable) or MTSET (membrane impermeable) then alkylated at

*Figure 2 continued on next page*

*Figure 2 continued*

unblocked cysteines. Control samples: (−) = not alkylated or blocked, (+) = alkylated but not blocked. % alkylation (%alk.) = high band/ total for each sample. The periplasmic localization score = (%alk.$_{MTSEA}$/% alk.$_{MTSET}$) and is indicated below the gel for each protein ± the 95% confidence interval from two biological replicates. Scores < 0.5 are cytoplasmic, near 1 are periplasmic.

The following figure supplement is available for figure 2:

**Figure supplement 1.** Growth rates of strains with different *cwlM* alleles.

## CwlM is phosphorylated by PknB

*Mtb* has 11 STPKs, which have been shown to phosphorylate many proteins involved in stress and growth (*Molle and Kremer, 2010*). To identify the cognate kinase of CwlM$_{TB}$, we performed phospho-transfer profiling (*Baer et al., 2014*) using proteins from the pathogen *Mtb*, which have high sequence conservation with the *Msmeg* homologues. We expressed and purified his-CwlM$_{TB}$ and his-MBP (Maltose Binding Protein) fusions to the *Mtb* kinase domains (KD) of PknA, B, D, E, F, H, J, K and L, and measured phosphorylation of his-CwlM$_{TB}$ using Western blotting with an α-phospho-threonine (α-thr~P) antibody. We found that his-MBP-PknB$_{TB}$(KD) phosphorylates his-CwlM$_{TB}$ more rapidly than the other kinases (*Figure 3A*; *Figure 3—figure supplement 1A,B*). We used mass spectrometry to show that *in vitro*, his-MBP-PknB$_{TB}$ phosphorylates his-CwlM$_{TB}$ at threonines 382, 384 and 386 (equivalent to T374, 376 and 378 in *M. smegmatis*, *Figure 3—figure supplement 1C*). Thus, the phosphorylation sites in the C-terminal tail of CwlM are conserved in *Mtb* and *Msmeg*, which supports the idea that the strong sequence conservation of the CwlM C-terminal tail across actinomyecetes (*Figure 2B*) is functionally relevant.

To test whether PknB affects CwlM phosphorylation *in vivo* we immunoprecipitated CwlM-FLAG from an *Msmeg* strain before and after induction of PknB$_{TB}$ overexpression, and determined the degree of phosphorylation by comparing reactivity with α-thr~P and α-FLAG antibodies (*Figure 3B*). We found that PknB$_{TB}$ overexpression increases CwlM phosphorylation. Thus, PknB is likely the primary kinase of CwlM. This is consistent with the known essential role of PknB in phosphorylating proteins involved in cell growth and division (*Fernandez et al., 2006*; *Kang, 2005*). PknE may also contribute to CwlM phosphorylation.

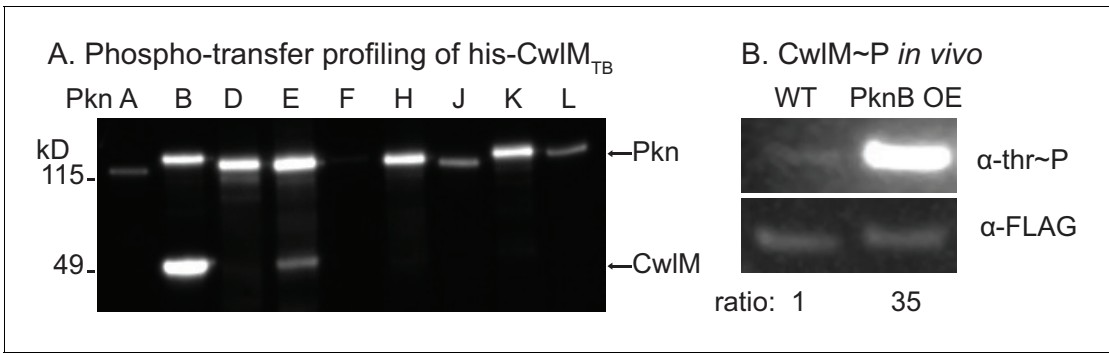

**Figure 3.** CwlM is phosphorylated by PknB. (A) α-thr~p western blots of *in vitro* kinase reactions with the kinase domains of 9 STPKs from *Mtb* fused to MBP and his-CwlM from *Mtb*. Reactions were stopped at 2 min. Longer time points are in *Figure 3-figure supplement 1B*. (B) α-thr~p and α-FLAG western blot of lysates from a strain expressing CwlM-FLAG in a PknB$_{TB}$ overexpression (CB418) background. PknB$_{TB}$ was uninduced or induced for 30 min with 100 ng/ml Atc. Ratio = signal of (α-thr~p / α-FLAG)$_{PknB\ OE}$ / (α-thr~p / α-FLAG)$_{uninduced}$. Experiments in (A) and (B) were both performed twice, representative images are shown.

The following figure supplement is available for figure 3:

**Figure supplement 1.** Phosphorylation of CwlM.

## CwlM interacts genetically and physically with MurA, a peptidoglycan precursor synthesis enzyme

Phosphorylated CwlM appears to be required for normal cellular growth. To identify the molecular role of CwlM, we screened for spontaneous suppressor mutants that grow rapidly despite carrying only the growth-restrictive, phospho-ablative *cwlM* T374A allele. We passaged this strain until we isolated rapidly growing mutants, which were mapped using whole genome sequencing.

Three of five suppressor strains contained an identical S368P mutation in the gene *murA*. MurA is a cytoplasmic enzyme that catalyzes the first dedicated step in PG precursor synthesis: transfer of an enolpyruvyl group from phosphoenol pyruvate (PEP) onto UDP-N-acetylglucosamine (UDP-GlcNAc)

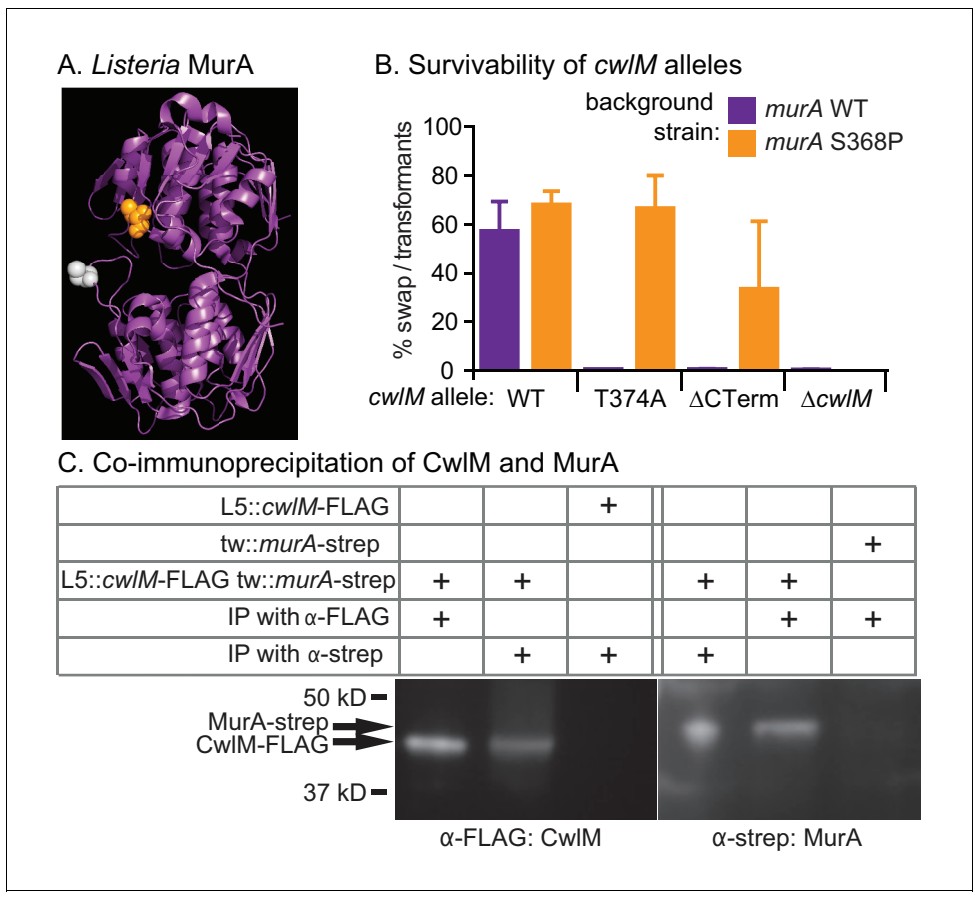

**Figure 4.** CwlM binds to and regulates MurA. (A) Crystal structure of MurA from *L. monocytogenes* (Halavaty et al., 2011). Orange: alanine that corresponds to S368 in *Msmeg* and *Mtb* MurA; gray: catalytic cysteine (aspartate in *Msmeg* and *Mtb*.). (B) Fitness of *cwlM* alleles in *murA* WT (CB737) and S368P (CB762) backgrounds, assessed by the percentage of colonies in which the WT *cwlM* allele was replaced by the indicated allele (WT = pCB277; T374A = pCB255; ΔCT = pCB557; Δ*cwlM* = pCB558). Four biological replicates of each transformation were performed, and 96–198 colonies were counted for each replicate. The Sidak correction for multiple comparisons was used to calculate p values using one-way ANOVA. The adjusted p values among the *murA* WT strains were calculated compared to *cwlM* WT: T374A =< 0.0001; ΔCT =< 0.0001; Δ*cwlM* =< 0.0001. The adjusted p values among the *murA* S368P strains were calculated compared to *cwlM* WT: T374A = 0.99; ΔCT = 0.006; Δ*cwlM* =< 0.0001. (C) α-FLAG and α-strep western blots of immunoprecipitates from the indicated strain (top three rows, from top: CB300, CB737, CB779) with the indicated antibody-conjugated beads (bottom two rows). The pulldown was performed twice, a representative image is shown.

The following figure supplement is available for figure 4:

**Figure supplement 1.** Growth rates of strains with different *cwlM* and *murA* regulatory alleles.

(*Marquardt et al., 1992*). The S368 residue is near the active site, according to an alignment with a structure of MurA from *L. monocytogenes* (*Halavaty et al., 2011*) (*Figure 4A*).

To confirm that the *murA* S368P allele suppresses the growth defect of the *cwlM* T374A allele in a different genetic background, we performed L5 allele swapping of *cwlM* in strains encoding either WT *murA* or *murA* S368P. In these strains a WT allele of *cwlM* is encoded on an L5-integrated construct marked with the nourseothricin resistance gene. We transformed these cells with kanamycin-marked L5 vectors with alternate alleles of *cwlM*. Appropriate recombination can be scored by gain of the kanamycin marker and loss of the nourseothricin marker. We found that the *cwlM* T374A allele and a deletion of the C-terminal tail are poorly tolerated in a WT *murA* background, but can be recovered at near WT levels in a *murA* S368P background. The entire *cwlM* gene cannot be deleted in either background (*Figure 4B*). The strain carrying *murA* S368P alone grows at the same rate as WT, the strain with both mutant alleles has a slight growth defect, but grows much more rapidly than the strain with *cwlM* T374A and WT *murA* (*Figure 4—figure supplement 1*).

These results confirm that *cwlM* and *murA* interact genetically. To test whether the proteins physically interact, we immunoprecipitated proteins from a strain containing *cwlM*-FLAG and *murA*-strep with both α-FLAG and α-strep beads. We found that CwlM-FLAG could be precipitated with α-strep beads, but only in the presence of MurA-strep, and vice versa (*Figure 4C*). Thus, there is a genetic link between *cwlM* and *murA,* and the two proteins interact.

## CwlM~P stimulates the enzymatic activity of MurA

We hypothesized that CwlM~P activates MurA and that MurA is less active when CwlM is less phosphorylated, resulting in the inhibition of cell growth seen in the *cwlM* T374A mutant. This hypothesis is based on the observation that the phospho-ablated *cwlM* T374A is suppressed by *murA* S368P (*Figure 4B*), which is likely to be a gain-of-function mutation because *murA* is an essential gene (*Griffin et al., 2011*) and the mutation does not affect growth rate (*Figure 4—figure supplement 1*). To test this model we expressed and purified his-MurA$_{TB}$ and his-CwlM$_{TB}$ from *E. coli* and performed kinetic assays of his-MurA$_{TB}$ activity by measuring the accumulation of the MurA product, enolpyruvyl-UDP-N-acetylglucosamine (EP-UDP-GlcNAc), by HPLC (*Figure 5A*). We compared the activity of his-MurA$_{TB}$ alone and in the presence of his-CwlM$_{TB}$ and his-CwlM$_{TB}$~P phosphorylated on T382, 384 and 386 by his-MBP-PknB$_{TB}$ (*Figure 3—figure supplement 1B*). We found that the reaction rate of his-MurA$_{TB}$ is 20–40 times faster in the presence of equimolar his-CwlM$_{TB}$~P (*Figure 5*) than it is alone or with unphosphorylated his-CwlM$_{TB}$. We found that nearly all his-CwlM$_{TB}$ incubated with his-MBP-PknB$_{TB}$ is phosphorylated on at least one site (*Figure 5—figure supplement 1F*); however, because there are three phosphorylation sites, it is likely a heterogeneous mixture. his-CwlM$_{TB}$ with different degrees of phosphorylation may stimulate MurA activity differently, thus the rates in *Figure 5D* represent a population average of the rates from his-CwlM$_{TB}$ in different states.

his-CwlM$_{TB}$ with phosphoablative mutations does not appreciably stimulate his-MurA$_{TB}$ activity after incubation with his-MBP-PknB$_{TB}$ and ATP (*Figure 5E*). his-CwlM$_{TB}$ alone cannot synthesize EP-UDP-GlcNAc; MurA is not thought to be phosphorylated (*Prisic et al., 2010*) and his-MBP-PknB$_{TB}$ cannot stimulate MurA activity in the absence of his-CwlM$_{TB}$ (*Figure 5E*). We measured the stoichiometry of the CwlM:MurA reaction and found that his-MurA$_{TB}$ is maximally activated in a reaction with a 2-fold molar excess of fully phosphorylated his-CwlM$_{TB}$~P (*Figure 5—figure supplement 1E, F*). Thus, CwlM~P activates MurA to initiate PG synthesis. These data with *Mtb* proteins confirm our model based on the genetic data in *Msmeg*, and show that CwlM$_{TB}$~P is a direct activator of MurA$_{TB}$.

## MurA activity is downregulated in the transition to starvation

MurA from other species have been studied *in vitro* and shown to have much more rapid kinetics and lower $K_m$ values than isolated MurA$_{TB}$ (*Figure 5B,C*; *Figure 5—figure supplement 1*; *Supplementary file 1D* [*Xu et al., 2014*]). Why would an essential housekeeping enzyme evolve to be almost inactive by default and require activation by another factor?

We hypothesized that this MurA regulatory pathway could exist to restrict PG synthesis in stresses such as nutrient starvation, because cell length decreases in starvation (*Figure 6—figure supplement 1A*) and when CwlM is deactivated (*Figures 1D, E*, *2E*). To test this, we immunoprecipitated CwlM-FLAG from *Msmeg* cultures in log phase and starvation and measured the degree of

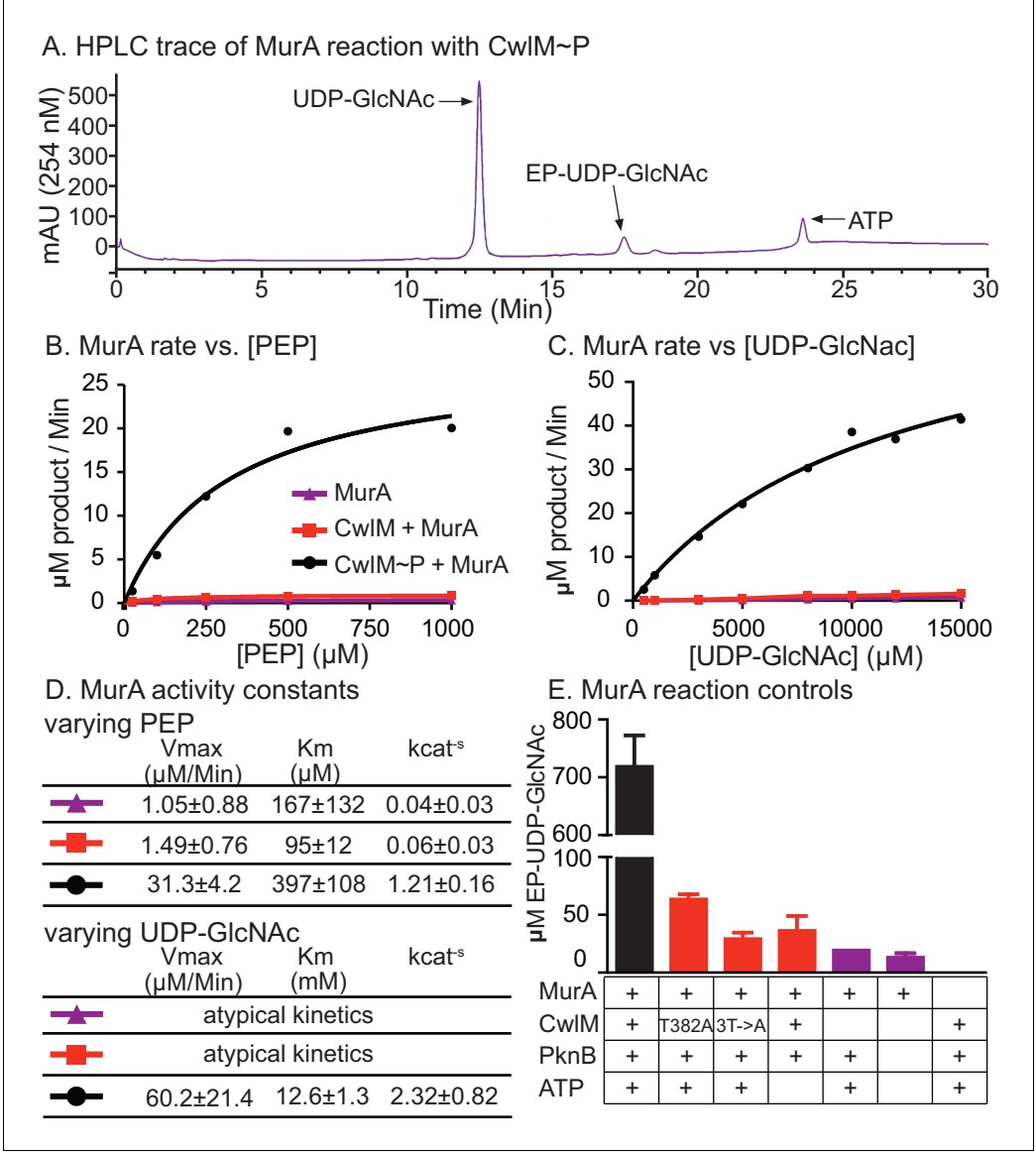

**Figure 5.** CwlM~P activates MurA *in vitro*. (**A**) An HPLC trace from the MurA$_{TB}$ kinetic assays, with substrate UDP-GlcNAc and product EP-UDP-GlcNAc indicated. Assays were done on his-MurA$_{TB}$ alone, his-MurA$_{TB}$ with equimolar his-CwlM$_{TB}$~P in an active kinase reaction (+ATP) with his-MBP-PknB$_{TB}$(KD) (a his-Maltose binding protein fusion to the kinase domain (KD) of PknB from *Mtb*), or his-MurA$_{TB}$ with his-CwlM$_{TB}$ in an inactive kinase reaction (-ATP) with his-MBP-PknB$_{TB}$(KD). (**B**) Rate of MurA activity vs. the concentration of phosphoenol pyruvate (PEP). Lines are the fit to the Michaelis Menten model. (**C**) Rate of MurA activity vs. the concentration of UDP-GlcNac for each MurA reaction condition. MurA and MurA+CwlM did not follow Michaelis Menten kinetics, (lines only connect the dots), for MurA+CwlM~P the line is the Michaelis Menten fit. The assays shown in (**B**) and (**C**) were done twice with two separate preps of his-MurA$_{TB}$ and his-CwlM$_{TB}$. Data from one replicate is shown here, full data in *Figure 5 - figure supplement 1A–D*. (**D**) Michaelis Menten kinetic constants from experiments in (**B**) and (**C**). Constants are the average ± the 95% confidence interval from the two replicates. (**E**) Amount of EP-UDP-GlcNAc produced after 30 min at 37°C in reactions with 2 mM PEP and 15 mM UDP-GlcNAc and his-MurA$_{TB}$ at 20 µg/ml; his-CwlM$_{TB}$ at 20.2 µg/ml; his-MBP-PknB$_{TB}$(KD) at 2 µg/ml and ATP at 1 mM where indicated. his-CwlM$_{TB}$ was incubated with his-MBP-PknB$_{TB}$ and ATP for 1 hr at RT before initiating the MurA reaction. T382A indicates that the singly-phosphoablative mutant protein his-CwlM$_{TB}$ T382A was used. 3T->A indicates that the triply-phosphoablative mutant protein his-CwlM$_{TB}$ T382A T384A T386A was used. Experiment was performed 2 times.

The following figure supplement is available for figure 5:

**Figure supplement 1.** MurA enzyme kinetics.

phosphorylation. We found that CwlM phosphorylation is quickly reduced upon starvation in PBS (*Figure 6A*).

These results, together with the biochemical data and the observation that MurA protein levels do not change substantially in starvation (*Figure 6—figure supplement 1B*), suggest that phosphorylation of CwlM converts MurA to an 'on' state and that during starvation, decreased CwlM phosphorylation blocks further PG synthesis - because MurA alone or in the presence of unphosphorylated CwlM has very low activity (*Figure 5*). To assess the importance of this post-translational regulation of MurA on mycobacterial physiology, we further studied the *murA* S368P mutant.

The genetic data predict that *murA* S368P is active without CwlM phosphorylation. To test this, we expressed and purified his-MurA$_{MS}$ WT and S368P and measured their activity alone and with his-CwlM$_{TB}$ and his-CwlM$_{TB}$~P. We found that his-MurA$_{MS}$S368P is more active than the WT protein in all conditions by about 2-fold (*Figure 6B*). This suggests that the mutant enzyme should produce more PG precursors than the WT strain, even in starvation. We assessed PG activity using a fluorescently labeled D-alanine analogue TAMRA-D-alanine (TADA), which is thought to be incorporated primarily into metabolically active PG (*Kuru et al., 2012*). We observed increased fluorescence in the strain carrying *murA* S368P as compared to the WT allele during the first 12 hr of PBS starvation, but, by 24 hr, the staining was equivalent between the strains (*Figure 6C*). Controls show that TADA staining is specific to the cell wall (*Figure 6—figure supplement 1C,D*). We interpret this to mean that post-translational regulation of MurA is important to control PG metabolism in logarithmic phase growth and during the transition to starvation, but that after prolonged starvation other regulatory mechanisms predominate to control PG metabolism. The abrupt cessation of growth (*Figure 6—figure supplement 1E*) and decreased CwlM phosphorylation (*Figure 6A*) imply that incorporation of new PG decreases upon starvation in the WT strain; however, TADA staining remains constant (*Figure 6C*). This suggests that TADA staining in these conditions may be due more to exchange of D-alanine in the periplasm by Ldts (*Cava et al., 2011*) than to incorporation of new, labeled PG transported from the cytoplasm. Regardless, the over-activation of MurA clearly alters PG metabolism during the transition to starvation.

## Regulation of PG biosynthesis contributes to antibiotic tolerance

Decreased synthesis of PG precursors and cell wall metabolism is an important adaptation to stress: subverting this regulation should be disadvantageous. Many models suggest that decreased metabolism contributes to Mtb's antibiotic tolerance during treatment. This is based on the observation that many antibiotics are less effective during starvation and stationary phase in Mtb (*Betts et al., 2002*; *Herbert et al., 1996*; *Wallis et al., 1999*; *Xie et al., 2005*) and other bacteria, likely because the activity of drug targets is reduced, or because permeability is decreased (*Kester and Fortune, 2014*; *Sarathy et al., 2013*). We hypothesized that continued synthesis of PG during the transition to starvation could reduce antibiotic tolerance. To test this, we subjected cells to PBS starvation and measured antibiotic sensitivity in strains with *murA* WT and S368P alleles. We found a 3–100 fold increase in killing of the strain carrying the *murA* S368P allele when the strains were treated with isoniazid, meropenem or rifampin at the initiation of starvation (*Figure 6D*). The mutant also had a significant antibiotic tolerance defect in nutrient rich media, but not when the antibiotics were added late in starvation (*Figure 6—figure supplement 1F,G*), after the period in which post-translational regulation of MurA affects PG metabolism (*Figure 6C*). We conclude that the PknB-CwlM-MurA signaling cascade functions to quickly turn off PG synthesis during nutrient restriction (*Figure 7*), and that this regulation contributes to antibiotic tolerance during growth and under changing conditions.

## CwlM protein evolution was driven by cytoplasmic localization

CwlM's function could not have been predicted from available knowledge about the function of its conserved domains. Based on domain structure, it seems likely that an ancestral *cwlM* homologue was a periplasmic enzyme involved in PG hydrolysis. How could such a protein evolve into a cytoplasmic, phosphorylation-responsive regulator of PG biosynthesis? To address this, we searched for *cwlM* homologues that retained both PG binding domains and the PG amidase domain in the same arrangement as is seen in the mycobacterial proteins (*Figure 7A*), and found that almost all such homologues were found in members of the phylum Actinobacteria. We chose representatives

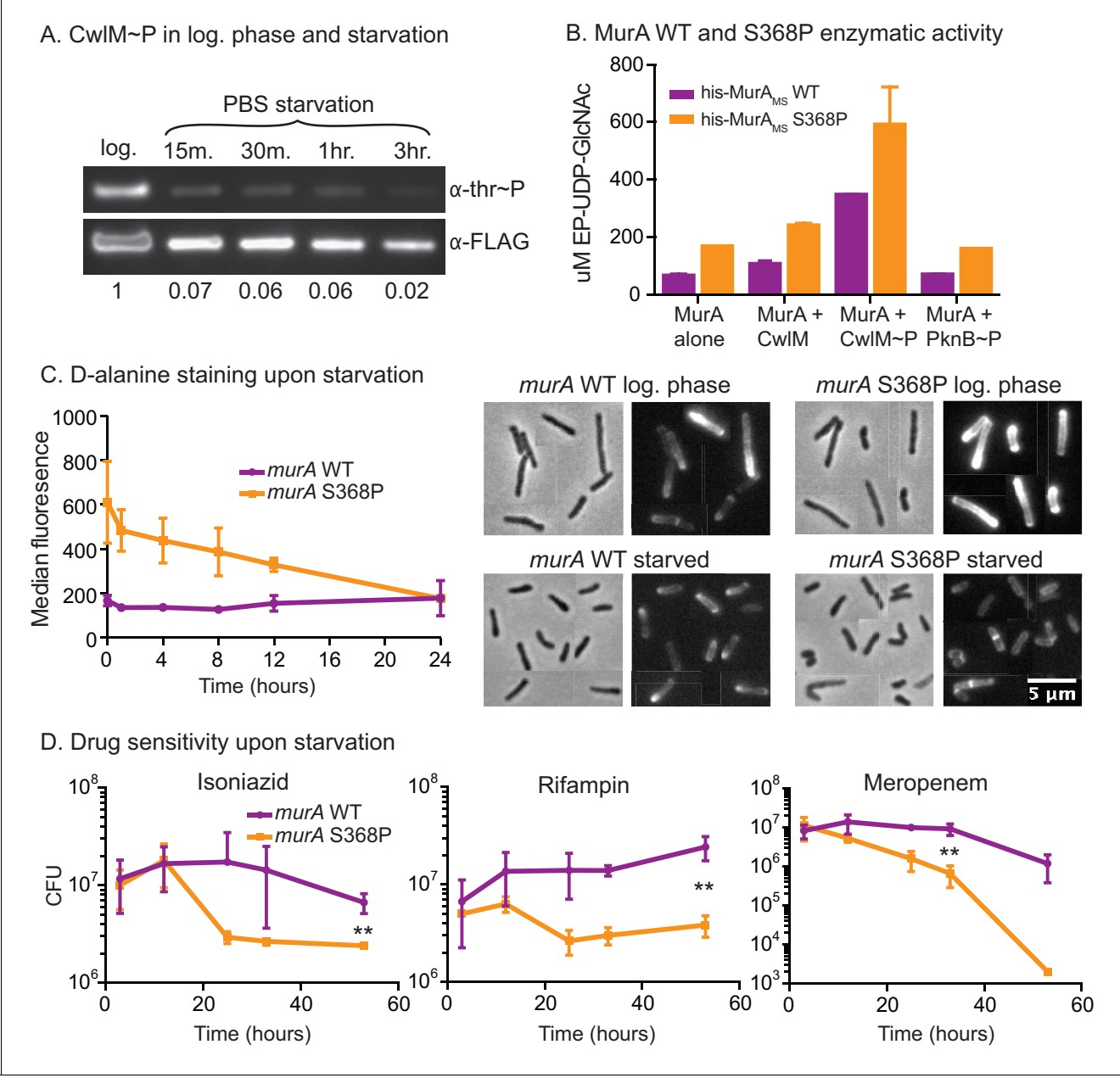

**Figure 6.** MurA is downregulated in starvation, contributing to drug tolerance. (**A**) α-thr~P western blot showing the level of CwlM phosphorylation in the *cwlM*::FLAG strain (CB100) in log. phase and upon starvation in PBS. Ratio = signal of (α-thr~p / α-FLAG)$_{starvation}$ / (α-thr~p / α-FLAG)$_{log\ phase}$(**B**) Amount of EP-UDP-GlcNAc produced after 30 min in a reaction with 20 µg/ml of his-MurA$_{MS}$WT or his-MurA$_{MS}$S368P, 2 mM PEP, 15 mM UDP-GlcNAc and either no or equimolar his-CwlM$_{TB}$ or his-CwlM$_{TB}$~P. (**C**) Fluorescent intensity of WT (CB779) and *murA* S368P (CB782) cells pulse stained with TADA in log. phase HdB culture (t = 0) and at time points after initiation of starvation in PBS+tween 80, as measured by flow cytometry. The median fluorescence for 10K+ cells is shown. On the right are representative images of cells from the t = 0 (log. phase) and t = 24 (starved) time points. Phase and red fluorescent images of each are shown. The images were taken with identical exposure settings and the brightness and contrast was adjusted to identical levels. Pictures of several cells from images processed identically were pasted together. The scale bar applies to all the images. (**D**) Colony forming units per ml of strains with the *murA* WT (CB779) and S368P (CB782) alleles after transfer to PBS+tween80 starvation media and treatment with 100 µg/ml isoniazid, 20 µg/ml meropenem or 50 ug/ml rifampin. The p value for the indicated time point in each experiment was calculated by the student's t test. Isoniazid = 0.0087; meropenem = 0.0082; rifampin = 0.0067.

The following figure supplement is available for figure 6:

**Figure supplement 1.** Phenotypes of *murA* S368P.

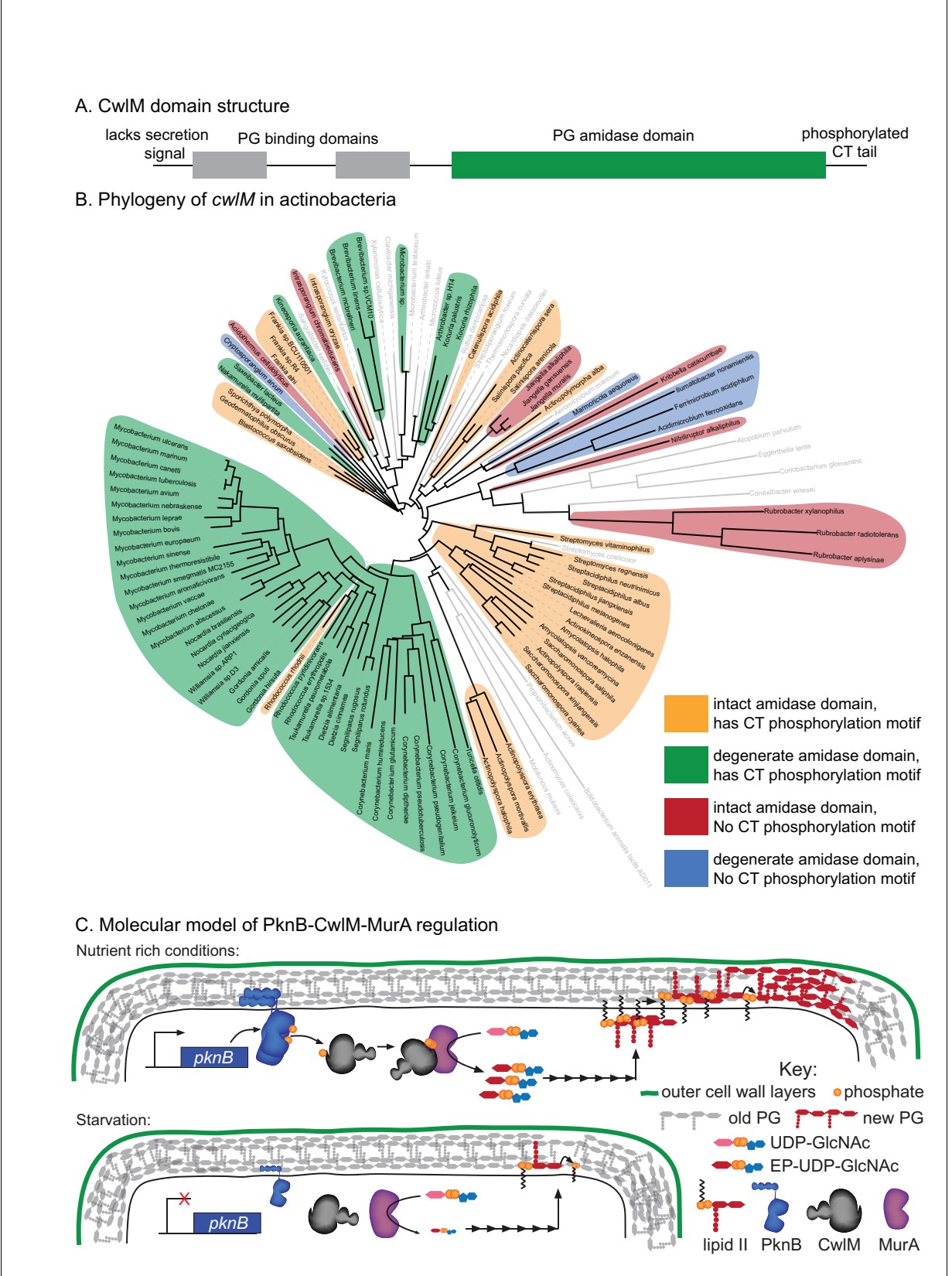

**Figure 7.** Phylogenetics and model for the PknB-CwlM-MurA regulatory pathway. (**A**) Domain structure of CwlM from mycobacteria. (**B**) Phylogenetic analysis of *cwlM* in the actinobacteria. Tree is based on a 16S rRNA alignment for each species. Species indicated in gray do not contain *cwlM*

*Figure 7 continued on next page*

Figure 7 continued

homologues, and are representatives of larger clades. All homologues lack a predicted secretion signal. The conservation of an intact or degenerate amidase active site and the presence of a PTG or TGT phosphorylation motif are indicated by color. (C) Model for the regulation of MurA. In nutrient rich conditions PknB phosphorylates CwlM, which can then promote the activity of MurA, resulting in increased flux through the PG biosynthetic pathway (indicated by the series of arrows) and promoting cell growth. In starvation, PknB is poorly expressed, CwlM is not phosphorylated, and MurA is therefore not activated, resulting in reduced production of PG precursors, which helps to halt cell growth.

from each family and searched for the presence of a secretion signal, the conservation of the four $Zn^{2+}$-coordinating residues, and the presence of a PTG or TGT phosphorylation motif in the C--terminal tail. We constructed a phylogenetic tree based on 16S rRNA sequence of the species that have *cwlM* homologues and some representative species from families that do not, and mapped onto it the conservation of the three *cwlM* features (*Figure 7B*). We found that all actinobacterial *cwlM* homologues lack secretion signals. Potential phosphorylation motifs are conserved in most clades, irrespective of whether or not they conserve the canonical $Zn^2+$-coordinating residues. These data imply that both CwlM cytoplasmic localization and, possibly, regulation by phosphorylation were present in an ancestor of the Actinobacteria.

## Discussion

Here we describe a regulatory system in which the essential STPK PknB phosphorylates the cytoplasmic PG amidase homologue CwlM, which then directly activates the catalytic activity of MurA, the first enzyme in the pathway of PG biosynthesis. During starvation, phosphorylation of CwlM is rapidly reduced, which leads to MurA deactivation and helps control PG metabolism at the cell wall. When this regulatory pathway is subverted by over-activation of MurA in *Msmeg*, cells become more sensitive to antibiotics.

### CwlM has a critical role in signal transduction

CwlM is essential in *M. smegmatis*, but its amidase activity does not appear to be essential (*Figure 1*). However, a strain with a phospho-ablative mutation, *cwlM* T374A, exhibits defects in elongation (*Figure 2*) and reduced viability in a manner reminiscent of CwlM depletion. The *cwlM* T374A mutant strain has such low viability that it could not be constructed except in a genetic background that 'forced' the L5 allele swap by repressing transcription of *cwlM* in transformants that maintained a wild type copy of the *cwlM* allele (See *Figure 2* and *4B* ; *Figure 4—figure supplement 1*; *Supplementary file 1* ). We show that the *murA* S368P mutation restores normal growth of the *cwlM* T374A mutant (*Figure 4B*; *Figure 4—figure supplement 1*) which implies that phosphorylation of CwlM activates MurA. The short cells seen in the CwlM depletion and the *cwlM* T374A mutant look similar to *Msmeg* cells with a temperature sensitive MurA mutation, which are also short and have compromised cell walls (*Xu et al., 2014*).

The loss of viability in the *cwlM* mutants and temperature sensitive *murA* strain (*Xu et al., 2014*) could be due to uncoordinated synthesis of the various layers of the cell wall: peptidoglycan biosynthesis is downregulated but other cell wall biosynthetic and metabolic processes may remain active, leading to a cell wall with poor structural integrity. This differs from wild-type regulation during starvation, when PG regulation is likely accompanied by regulation of other cell wall layers. In fact, PknB, the kinase likely responsible for physiologic CwlM phosphorylation, has been implicated in the regulation of many cell wall processes (*Khan et al., 2010*; *Molle et al., 2006*; *Parikh et al., 2009*; *Vilchèze et al., 2014*), and probably helps to coordinate regulation of the various cell wall layers. The enzymatic activity of other PknB substrates changes less upon phosphorylation (*Khan et al., 2010*; *Molle et al., 2006*; *Parikh et al., 2009*) than the ~30-fold activation of MurA by CwlM~P. It is possible that PknB may regulate other factors like it does MurA – indirectly through intermediary substrates.

### CwlM may integrate information from multiple signals

Both PknB and PknE can phosphorylate CwlM *in vitro*. Because PknB acts more rapidly (*Figure 3A*) it is likely to be the primary kinase of CwlM. In nutrient replete conditions PknB apparently activates

MurA by phosphorylating CwlM. How is the phosphorylation cascade regulated in starvation? PknB can be transcriptionally (*Kang, 2005*) and post-translationally regulated (*Baer et al., 2014*). However, the rapid decrease in CwlM phosphorylation (*Figure 6B*) implies that there is a CwlM phosphatase.

CwlM is a cytoplasmic PG hydrolase homologue. While most PG hydrolases are in the periplasm, there are cytoplasmic PG hydrolases that are involved in enzymatic processing of recycled anhydro-muropeptides (*Jacobs et al., 1994*; *1995*; *Park and Uehara, 2008*). The level of cytoplasmic anhydro-muropeptides can be a signal reporting on environmental conditions (*Jacobs et al., 1994*; *1995*; *Park and Uehara, 2008;Núñez et al., 2000*). If CwlM is involved in processing or detection of anhydro-muropeptides, it could integrate information about their levels into its regulation of PG precursor synthesis.

## Regulation of MurA toggles PG biosynthesis in response to prevailing conditions

Because MurA is the first committed step in PG biosynthesis, its regulation is an efficient point at which to regulate the pathway. This is certainly true in other species. MurA from *E. coli* is subject to feedback regulation (*Mizyed et al., 2005*), and many Gram positive species have two MurA homologues which are differentially regulated (*Blake et al., 2009*; *Kock et al., 2003*). Mycobacteria have only one MurA homologue, but the regulation by CwlM allows it to quickly and precisely adjust its enzymatic activity for different conditions without being proteolyzed and requiring re-synthesis (*Figure 6—figure supplement 1B*). Our data imply that the post-translational regulation of MurA is important for cell wall regulation at early time points during stress (*Figure 6C*). It is likely that transcriptional (*Dahl et al., 2003*) and proteolytic (*Festa et al., 2010*) systems downregulate cell wall enzymes after prolonged starvation or stress.

The $K_{cat}$ values we measured for MurA alone by varying PEP concentration are comparable to the values measured previously (*Xu et al., 2014*), and the $K_{cat}$ of MurA with CwlM~P was ~30 times higher. We and others have measured high $K_m$ values for UDP-GlcNAc (*Figure 5D*; *Supplementary file 1D*; *Figure 5—figure supplement 1* and [*Xu et al., 2014*]), which surely exceed the concentrations in the cell; thus, there are probably other factors that regulate MurA and either reduce the $K_m$ or increase the local concentration of UDP-GlcNAc.

## A regulatory system for PG biosynthesis contributes to drug tolerance

One of the major limitations in treating tuberculosis is the extended course of therapy required in part because of phenotypic drug tolerance, which is thought to be a result of decreased metabolic activity of drug targets, and decreased permeability. Subverting signals that downregulate metabolism in these bacteria could re-sensitize them to antibiotics. We find that this is true during the transition to starvation in *Msmeg* grown in culture. A strain carrying the *murA* S368P allele is unable to properly downregulate PG metabolism, and is more sensitive to several antibiotics early in starvation (*Figure 6D*). Because this increased killing was seen in the presence of isoniazid and meropenem, which target the cell wall, as well as rifampin, which does not, we think that the increased cell wall metabolism results in higher permeability, and that the increased killing is due to greater drug uptake (*Sarathy et al., 2013*).

Our data imply that MurA inhibitors (*De Smet et al., 1999*) might contribute to death of actively growing mycobacteria early in treatment, but that they could also contribute to drug tolerance of non-growing bacteria, which are likely to dominate later in treatment. While our experiments were conducted *in vitro* with *Mtb* proteins, the conservation of these proteins among mycobacteria suggest that the PknB-CwlM-MurA regulatory pathway is conserved, although it is likely that different stresses activate MurA regulation in *Mtb* compared to *Msmeg*. Treatments that interfere with the regulation of the mycobacterial cell wall during infection could shorten treatment times and improve patient outcomes.

## The evolution of CwlM function

Biological complexity is predicated on the diversity of protein function. How do novel protein functions evolve? One hypothesis is that compartmental re-targeting of duplicated genes is critical to

the evolution of complexity in eukaryotic cells (*Bright et al., 2010*; *Gabaldón and Pittis, 2015*). It is less clear that this same mechanism would drive the evolution of complexity in bacteria.

Here, though, we find a bacterial example of evolutionary repurposing through new compartmentalization. Because CwlM consists of three functional domains, all of which are predicted to function in the periplasm, we assume that an ancestral homologue was periplasmic. However, all of the available *cwlM* homologues lack a predicted secretion signal. These homologues are found throughout the Actinobacteria, implying that a *cwlM* ancestral homologue was re-targeted to the cytoplasm before the actinobacterial phylum split off from its relatives. The conservation of the putative phosphorylation motif implies that *cwlM* homologues in most Actinobacteria have a role in signal transduction. Because periplasmic proteins are not known to be phosphorylated, this signaling role likely evolved after CwlM was re-targeted to the cytoplasm.

CwlM provides cells with an additional layer of regulation of PG synthesis. MurA from Gram negative (*Dai et al., 2002*; *Krekel et al., 2000*) and positive species (*Blake et al., 2009*; *Du et al., 2000*) are regulated transcriptionally and by proteolysis (*Blake et al., 2009*; *Kock et al., 2003*). In mycobacteria, enzymes in PG biosynthesis are regulated transcriptionally (*Dahl et al., 2003*) and proteolytically (*Festa et al., 2010*), in addition to the regulation of MurA through phosphorylation of CwlM described here. Thus, compartmental re-targeting of the ancestral CwlM protein has allowed for the evolution of a complex regulatory system by which mycobacteria, and possibly related Actinobacteria, control synthesis of PG precursors in response to environmental conditions.

## Materials and methods

### Bacterial strains and culture conditions

*Mycobacterium smegmatis* mc$^2$155 was grown in 7H9 salts (Becton-Dickinson, Franklin Lakes, NJ) supplemented with: 5 g/L albumin, 2 g/L dextrose, 0.85 g/L NaCl, 0.003 g/L catalase, 0.2% glycerol and 0.05% Tween80, or plated on LB agar. Hartmans-de Bont (HdB) media was made as described (*Hartmans and De Bont, 1992*) with 0.05% tween80, HdB-C was made without glycerol. *E. coli* DH5α was used for cloning and *E. coli* BL21 codon plus or ArcticExpress DE3 RP were used for protein expression. Antibiotic concentrations for *M. smegmatis* were: 25 µg/ml kanamycin, 50 µg/ml hygromycin, 20 µg/ml zeocin and 20 µg/ml nourseothricin. Antibiotic concentrations of *E. coli* were: 50 µg/ml kanamycin, 100 µg/ml hygromycin, 25 µg/ml zeocin, 40 µg/ml nourseothricin, 20 µg/ml chloramphenicol and 140 µg/ml ampicillin. All strains were grown at 37°C.

### Strain construction

Knockouts of *cwlM* and *murA* were made by first complementing the genes at the phage integrase sites L5 (*Lewis and Hatfull, 2003*) and Tweety (*Pham et al., 2007*), and then using recombineering to delete the endogenous copy. ~500 base pair regions upstream and downstream of the gene were amplified by PCR and PCR stitched to either side of a zeocin resistance cassette. The assembled PCR fragment was transformed into *Msmeg* strains expressing the RecET proteins. The resulting colonies were screened for deletion of the gene (*van Kessel and Hatfull, 2008*). To make CB737, the zeocin resistance cassette in the deletions was removed by the Cre recombinase between each subsequent deletion. After complementing *murA* at the Tweety site and deleting the endogenous *murA*, we found that there was still another copy of *murA* in its native genomic context. This led to our discovery of an IS1096-mediated (*Wang et al., 2008*) genomic duplication which comprises MSMEG_4928–4944 (*murA* is MSMEG_4932) and is present in most of our *Msmeg* strains. We deleted the second copy of *murA* by removing the zeocin cassette in the first deletion via Cre recombinase, and using the same knockout construct to delete the second copy.

The *cwlM*::FLAG (CB100) strain was also made by recombineering. ~500 base pair regions upstream and downstream of the stop codon of *cwlM* were amplified and PCR stitched to a zeocin cassette, which was transformed as described above.

Different alleles of *cwlM* were attained by swapping the nourseothricin resistance-marked vector with WT *cwlM* for a kanamycin resistance-marked mutant allele, as described (*Pashley and Parish, 2003*). We also exchanged the *murA* allele at the Tweety integrase site in the same way as is done at the L5 site. Full strain details in *Supplementary file 1A–C*.

## Cell staining, microscopy and image analysis

For polar growth quantification, cells were stained with AlexaFluor488 carboxylic acid succinimidyl ester as described (*Aldridge et al., 2012*), resuspended in 7H9 media, shook at 37°C for 3 hr, immobilized on agarose pads and imaged in phase and GFP channels with a Nikon TE-200E microscope with a 60X objective and Orca-II CCD camera (Hamamatsu, Japan). The Ptet::*cwlM* strain was grown uninduced for 9 hr before staining. Images were analyzed in ImageJ (NIH). The length of the longer unstained pole and the total cell length was measured manually for each cell.

## D-alanine staining and flow cytometry

For the TADA experiment, cells were grown in HdB. 1 µl of 10 mM TADA (synthesized according to [*Kuru et al., 2014*]) was added to 1 mL of culture and incubated for 5 min before washing with PBS +tween80 and fixing for 10 min with 1% paraformaldehyde. The rest of the culture was pelleted, resuspended and cultured in PBS+tween80, and aliquots were stained the same way at 1, 4, 8, 12 and 24 hr after resuspension. Controls were performed on cells growing in HdB: cells were either fixed first and then stained, or fixed and not stained at all. The samples were filtered through a 10 µm filter and analyzed by flow cytometry (MACSQuant VYB Excitation: 561 nm; Emission filter: 615/ 20). The settings for FSC, SSC, and cell density were adjusted to sample single cells. Three biological replicates were used for each strain and time point, and the median fluorescence for 50,000 cells from each replicate was calculated and averaged.

## D-alanine staining and cell fractionation

75 mL replicate cultures of *murA* WT and *murA* S368P growing in HdB were stained for 15 min at 37°C with 10 µM TADA. Cells were pelleted and washed twice with PBS+tween80, resuspended in water and boiled for 1 hr, and spun at 100 K rpm for 20 min. The supernatants were collected, and the pellets resuspended in PBS+2.5% SDS, boiled for 1 hr, nutated O/N at 37°C, and spun at 21 K rpm for 15 min. The supernatants were collected and the pellets were washed twice with PBS. The fluorescence of each sample was measured with a Spectra284 plate reader (Excitation: 541, Emission: 568). The water and SDS extracted supernatants were measured undiluted. The cell wall pellet suspension was diluted 120 fold, which was required for measurements to be within the linear range. The fluorescence values of the cell wall fractions measured by the plate reader were therefore multiplied by 120.

## Protein purification and phospho-transfer profiling

N-terminally his-MBP-tagged kinase domains of the nine canonical serine-threonine protein kinases from Mtb were expressed and purified as described (*Kieser et al., 2015*). His-CwlM was expressed with 1 mM IPTG at 14° for 40 hr in ArcticExpress DE3 RP (Agilent, Lexington, MA). Cells were resuspended and French-pressed in Ni Wash Buffer (50 mM NaHPO$_4$ pH 8.0, 300 mM NaCl, 20 mM imidazole), and supernatants were poured over TALON affinity resin (Clontech, Mountain View, CA). Bound proteins were washed and eluted with Ni Wash Buffer + 200 mM imidazole. Soluble proteins were separated from aggregates on a Superdex S200 gel filtration column (GE Healthcare, Westborough, MA) in 20 mM Tris pH 7.5, 150 mM NaCl, 1 mM DTT. Soluble proteins were concentrated and stored in 50 mM NaPO$_4$ pH 7.5, 150 mM NaCl, 20% glycerol, 2 mM DTT, 1 mM PMSF. Kinase reactions, α-phospho-threonine western blots and mass spectrometry were performed as described (*Kieser et al., 2015*).

## Growth rate determination

Growth curves of *M. smegmatis* strains were done in triplicate, the OD$_{600}$ of each culture was measured every 30 min in a Bioscreen growth curve machine (Growth Curves USA, Piscataway, NJ).

## Conservation of C-terminal tail

To determine the conserved features of the C-terminus of CwlM, the entire CwlM protein sequence was BLASTed against the genus Mycobacteria, and the C-terminal tail from the 34 most similar mycobacterial, and 98 most similar Actinomycetes CwlM proteins was analyzed using Weblogo (*Crooks et al., 2004*).

## 2D gels, SDS-PAGE and western blots

For 2D gels and phosphothreonine quantification of CwlM, CwlM-FLAG was immunoprecipitated by bead beating cells in 50 mM Tris pH7.5, 300 mM NaCl with Protease Inhibitor Cocktail (Roche, Switzerland). $\alpha$-FLAG M2 Magnetic Beads (Sigma Aldrich, Natick, MA) were added to supernatants and washed with 50 mM NaHPO$_4$, 300 mM NaCl. CwlM was eluted from the beads with 0.5 mg/ml FLAG peptide in TBS. Samples were separated in 2D using the ReadyPrep 2D Starter Kit (Bio-Rad, Hercules, CA), according to the manufacturer's protocols. SDS-PAGE was done with 4–12% NuPAGE Bis Tris precast gels (Life Technologies, Beverley, MA). Mouse $\alpha$-FLAG (Sigma Aldrich) was used at 1:10,000 in TTBS, Rabbit $\alpha$-strep (Genscript, China) and Rabbit $\alpha$-phosphothreonine (Cell Signaling Technology, Danvers, MA) were used at 1:1000 in TTBS + 0.5% BSA.

## Substituted cysteine accessibility

Cultures of the L5::cwlM1cys-strep, L5::lprG-strepC and L5::pgm-strepC strains were grown to late log phase, washed twice and resuspended in PBS + 0.25% tween80, split and treated with 0 or 0.6 mg/ml of MTSEA and MTSET (Biotium, Hayward, CA) for 20 min at RT. Blocking was quenched with cysteine at 50 mM; cells washed in PBS + 0.05% tween80, and resuspended in PEGylation buffer (600 mM Tris pH 7.4, 10 M Urea, 2% SDS, 1 mM EDTA), heated at 85° for 20 min and spun. 5.6 mg of MalPEG5K (Sigma Aldrich) was added to all the supernatants except (-) controls and incubated at 37° for 2 hr. Proteins were TCA precipitated and visualized by $\alpha$-strep western blot.

## CwlM allele survivability

To confirm the suppression of the cwlM T374A allele by the murA S368P allele, the L5::cwlM(nuoR) tw::murA (CB737) and L5::cwlM(nuoR) tw::murA S368P (CB762) strains were transformed with pCB255, pCB277, pCB557 and pCB558 and plated on kanamycin. For each of 3–4 transformations, kanamycin and nourseothricin resistance was assessed for 96-192 colonies.

## CwlM-MurA co-immunoprecipitation

Cultures of CB779, CB737 and CB300 were grown to mid-log phase, washed, resuspended in PBS + 0.25% paraformaldehyde, incubated at 37° for 2 hr, quenched with 40 mM glycine, washed and lysed by French press. $\alpha$-strep was conjugated to Protein G dynabeads (Novex, Cambridge, MA) which were added to the CB300 and the CB779 lysate. $\alpha$-FLAG beads were added to the rest of the CB779 lysate and the CB737 lysate. Pull-downs were done according to the manufacturers protocols, and proteins visualized by western blot.

## MurA kinetic assays

his-MurA was purified like his-CwlM. Reactions with 20 µg/ml of his-MurA, equimolar his-CwlM in the kinase reaction, and varying concentrations of substrate were run as in (*Brown et al., 1994*). MurA kinetic assays were done with MurA$_{TB}$ alone in MurA reaction buffer (50 mM Tris pH 8.0, 2 mM Kcl, 2 mM DTT) or with equimolar CwlM$_{TB}$ in a kinase reaction with His-MBP-PknB$_{KD}$. The kinase reactions contained: 202 µg/ml CwlM$_{TB}$, 50 µg/ml His-MBP-PknB$_{KD}$, 2 mM ATP, 2 mM MnCl$_2$ and were brought up to 100 µl with Buffer C (20 mM Tris pH 7.5, 150 mM NaCl, 1 mM DTT). Inactive kinase reactions had extra buffer instead of ATP. Kinase reactions were run for 1 hr at room temperature before being added to the MurA reactions. MurA reactions contained: 20 µg/ml MurA and 20.2 µg/ml CwlM in the kinase reaction (1/10 of MurA reaction was CwlM kinase reaction, or buffer for MurA alone kinetics). In MurA reactions with varying PEP, UDP-GlcNac was at 10 mM and PEP concentrations were: 25, 100, 250, 500 and 1000 µM; these reactions were started with the addition of UDP-GlcNAc. In MurA reaction with varying UDP-GlcNac, PEP was at 2 mM and UDP GlcNac concentrations were: 0.5, 1, 3, 5, 8, 10 and 15 mM; these reactions were started with the addition of PEP. 2, 5 and 8 min time points were taken for the MurA + CwlM~P reaction; 5, 15 and 25 min time points were taken for the MurA and MurA +CwlM reactions. At the indicated time point, a 25 µl aliquot of the reaction was removed and quenched in an equal volume of 400 mM KOH. The quenched reactions were spun in Microcon 10 K devices (EMD Millipore, Billerica, MA) and chilled until being injected into the Agilent HPLC. HPLC separation was performed on a MonoQ 5/50 GL anion exchange column (GE Healthcare) with the following protocol: 0.6 ml/min; 2 min of 20 mM tetraethylammonium bicarbonate pH 8.0, an 18 min gradient from 20–500 mM tetraethylammonium

bicarbonate pH 8.0, 5 min of 500 mM tetraethylammonium bicarbonate pH 8.0, 5 min of 20 mM tetraethylammonium bicarbonate pH 8.0. The area under the $A_{254}$ curve peak corresponding to EP-UDP-GlcNAc was integrated using Agilent ChemStation software.

Because EP-UDP-GlcNac is not commercially available, we were not able to perform a standard curve of product concentrations to determine the relationship between peak area and product concentration. Instead, we measured the peak area of a range of UDP-GlcNac concentrations, and found that a peak area of 11.73 in the $A_{254}$ curve was consistently equivalent to 1 uM of UDP-GlcNac. Because UDP-GlcNac and EP-UDP-GlcNac differ only in one enol-pyruvyl group, we reasoned that the relationship between $A_{254}$ peak area and concentration should be comparable between these two molecules. We therefore divided the values for peak area for each EP-UDP-GlcNac peak by 11.73 to calculate the approximate µM concentration of EP-UDP-GlcNac. These values were plotted vs. time for each substrate concentration and the rate of product produced vs. time was calculated based on the linear portions of the curves. The resulting rates were plotted against substrate concentration in the curves shown in *Figure 5* and *Figure 5—figure supplement 1*. The peak that corresponds to EP-UDP-GlcNAc was identified because it appeared only when MurA and UDP-GlcNAc were incubated with PEP. The data were fitted to the Michaelis Menten formula using GraphPad Prism.

## Antibiotic kill curves

Biological replicates of CB779 (*murA*) and CB782 (*murA S368P*) were grown in 7H9, transferred to PBS+tween80, diluted to OD = 0.05, treated with 100 µg/ml isoniazid, 50 µg/ml rifampin or 20 µg/ml meropenem, and rolled at 37° during the CFU measurement period. For the unstarved kill curves (*Figure 6—figure supplement 1C*), log. phase cultures in 7H9 were diluted to OD = 0.05 in 7H9 and treated with 50 µg/ml isoniazid, 20 µg/ml rifampin or 5 µg/ml meropenem. For *Figure 6—figure supplement 1D*, the log. phase 7H9 cultures were transferred to PBS+tween80, diluted to OD = 0.05 and rolled at 37° for 18 hr before 100 µg/ml isoniazid, 50 µg/ml rifampin or 20 µg/ml meropenem was added.

## CwlM feature conservation analysis

A CDART search was used to identify *cwlM* homologues with the same domain structure. All but 3 of these were in the Actinobacteria, the remaining 3 are in the Firmicutes. The ~700 CDART hits were separated by family, and a phylogenetic tree was made of each family. Between 3 and 16 homologues, representing the breadth of the phylogenetic distribution, were chosen from each family, leaving 97 homologues. The secretion signal identification programs Phobius (*Kall et al., 2007*) and PRED-TAT (*Bagos et al., 2010*) were used to assess the N-terminus of each homologue for the presence of a Sec or Tat secretion signal. The COBALT tool on NCBI was used to align the amidase domain of each homologue to active amidase domains of the same family; the conservation of the four $Zn^{+2}$-coordinating residues was assessed for each. Each homologue was then searched for the presence of either PTG or TGT motifs in the C-terminal region after the amidase domain. 16S rRNA sequences were collected from the species with the 97 CwlM homologues, and from representative species from across the clade of Actinobacteria (*Lang et al., 2013*) that do not have CwlM homologues. The 16S sequences were aligned with Clustal Omega and a distance-matrix tree was constructed and visualized in ITOL (*Letunic and Bork, 2011*).

## Acknowledgements

We thank Tom Bernhardt for critical reading of the manuscript; Suzanne Walker, Tom Bernhardt and David Rudner for generosity with equipment; and Skye Fishbein for help with data processing. This work was supported by National Institutes of Health NIAID U19 AI107774 to EJR, and the Ruth L Kirschstein National Research Service Award 5F32AI109850-02 from NIAID to CCB.

## Additional information

### Funding

| Funder | Grant reference number | Author |
|---|---|---|
| NIH Office of the Director | U19 AI107774 | Eric J Rubin |
| NIH Office of the Director | 5F32AI109850-02 | Cara C Boutte |

The funders had no role in study design, data collection and interpretation, or the decision to submit the work for publication.

### Author contributions

CCB, Conceptualization, Methodology, Investigation, Resources, Writing-Original Draft, Writing-Review and Editing, Visualization, Funding acquisition, Acquisition of data, Analysis and interpretation of data, Contributed unpublished essential data or reagents; CEB, Methodology, Resources, Writing-Review and Editing, Contributed unpublished essential data or reagents; KP, Investigation, Resources, Acquisition of data, Drafting or revising the article, Contributed unpublished essential data or reagents; WL, Investigation, Acquisition of data; MRC, Methodology, Acquisition of data, Analysis and interpretation of data; XM, Investigation, Acquisition of data, Contributed unpublished essential data or reagents; SMF, Supervision; CMS, Resources, Writing-Review and Editing, Supervision, Contributed unpublished essential data or reagents; TRI, Investigation, Writing-Review and Editing, Acquisition of data, Analysis and interpretation of data; EJR, Conception and design, Analysis and interpretation of data, Drafting or revising the article

### Author ORCIDs

Cara C Boutte, http://orcid.org/0000-0001-7645-6221
Eric J Rubin, http://orcid.org/0000-0001-5120-962X

## Additional files

### Supplementary files

• Supplementary file 1. Strain information and table of MurA kinetics across species. (a) Strains. Strains used, with strain number, abbreviated name used in the text and full genotype. (b) Plasmids. Plasmids used, with original reference for parent plasmid, and strain cross-reference. (c) Primers. Primers used to construct each strain. Those beginning with 'p' are plasmids, others are recombineered alterations on the chromosome. (d) Kinetic parameters of MurA proteins pulled from the literature.

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
