## [Decision Letter]

Thank you for submitting your work entitled "A cytoplasmic peptidoglycan amidase homologue controls mycobacterial cell wall synthesis" for consideration by *eLife*. Your article has been reviewed by three peer reviewers, one of whom is a member of our Board of Reviewing Editors, and the evaluation has been overseen by Detlef Weigel as the Senior Editor.

The reviewers have discussed the reviews with one another and the Reviewing Editor has drafted this decision to help you prepare a revised submission.

Summary:

The reviewers liked several aspects of the work and found a possible role of CwlM as a cytoplasmic regulator of MurA in mycobacteria intriguing. However, there were also a number of concerns raised about the completeness and rigor of many of the experiments presented. The collective comments of the reviewers have been merged and are listed below in two categories, Essential Revisions and Minor Points. As you will see, the Essential Revisions include a number of cases where additional experiments will be required. If you feel that each of the items below can be fully addressed, we would encourage you to submit a revised manuscript that would then be re-reviewed.

Essential revisions:

1) The claim that CwlM is non-enzymatic is not sufficiently rigorous. It is true that mutations in putative catalytic residues do not cause a loss of function, but this is rather weak; they need to show explicitly that the mutant protein is inactive biochemically. In addition, the mutagenesis is based on the analysis of a truncated *B. subtilis* protein and the paper cited describes numerous mutations not in the catalytic residues that also inactivated the protein. So, the linkage between specific mutations and lack of catalysis is tenuous. Additionally, according to Deng et al. it does have activity – do you not agree and if so, why?

2) Figure 1: why does depleting CwlM cause cells to die rather than just stop growing? If the model is that CwlM is simply required for promoting cell elongation, then why don't cells lacking CwlM just stop growing? As shown later, when CwlM phosphorylation is downregulated in starvation conditions, cells stop growing, but presumably are not dying, so reconciling these data is important. Figure 1: Do CwlM-depleted cells phenocopy MurA-depleted cells (Xu et al., 2014) or show any shape changes or lysis?

3) Does the T374A mutant lose viability like the depletion strain? The doubling time is longer (Figure 2) and the cells don't grow polarly (Figure 2) but what about viability?

4) The authors cite Kall et al. 2007 as evidence that CwlM lacks a secretion signal, but that paper is only a citation to a prediction server and furthermore, running CwlM through that server yields the prediction of a non-cytoplasmic (i.e., secreted) protein. The determination of compartmental localization using the MTSEA/MTSET is also confusing and should be clarified. The authors state in the legend to Figure 2 that "scores >1 are periplasmic, <1 are cytoplasmic", so why does mRFP, unambiguously a cytoplasmic protein, have a value of.84 ±. 40. In addition to a better explanation of the MTSEA/MTSET approach, the authors should consider a more traditional approach, such as fractionation, unless it is experimentally too demanding. And, perhaps more importantly, is it possible that some CwlM is present in each compartment? Unless this can be categorically ruled out, the authors should adjust their language about the cellular location of CwlM throughout the paper, including in the title.

5) T374 is phosphorylated and the T374A mutant displays some of the phenotypes associated with a *cwlM* depletion strain. But formally such data alone do not support the conclusion that "CwlM is a cytoplasmic protein whose essential function is activated by phosphorylation." It could be well be that the threonine at position 374 is important for its essential function but not because it must be phosphorylated. At a bare minimum, the authors need to be more precise in their language. Preferably, they would address this experimentally – for example, by testing whether a phosphomimetic (T374D or T374E) allele is hyperactive with respect to activating MurA – has this been tried?

6) Do constitutively active forms of CwlM rescue the essentiality of *pknB* mutants? And does the S368P mutation in *murA* bypass essentiality of *pknB*?

7) Figure 5: The activation of MurA by CwlM+PknB is convincing, although the authors need to also show that MurA is not activated by CwlM(T347A)+PknB to ensure that PknB isn't having some other effect independent of T347 phosphorylation. Also, does CwlM(T347A) no longer activate MurA, as predicted from the genetics? Finally, is the basal level activity of the Mtb version of MurA(S368P) significantly higher than that of wt MurA, and/or is unphosphorylated CwlM sufficient to stimulate its activity?

8) In the experiment in Figure 5 showing the effect of phosphorylation on the ability of CwlM to activate MurA, they need to characterize the stoichiometry of phosphorylation – the data indicate that this is quite high, but that would be the exception for bacterial phosphoproteins. Also, as noted above, they should try the phosphomimetic mutant of T374 to confirm the genetic data from Figure 2.

9) Figure 6 suggests that the change in CwlM phosphorylation is because of a change in PknB levels, but the authors need to verify this by showing that constitutive expression of PknB maintains CwlM phosphorylation upon starvation.

10) Figure 6, subsection “MurA activity is downregulated in starvation” – The TADA incorporation experiments (Figure 6) suggest that the *murA*(S368P) cells still produce PG upon starvation when the wt cells do not (so much). The authors suggest that growth inhibitory mechanisms are still in effect because cells of both strains become short. Hence, increased TADA incorporation is probably correlated with increased PG turn-over, rather than increased net synthesis, in the *murA*(S368P) cells. The authors may be correct, but only if cell size is (still) a reliable marker of cell growth.

a) It is conceivable that the *murA*(S368P) mutant cells, though small, do keep on growing and dividing for longer (reach a higher OD) under starvation conditions. For example, depletion of some intermediate of LipidII synthesis might normally act as a general signal to stop/slow growth in wt cells, and this might no longer be very effective in the mutant. I recommend that the authors do some simple growth curves to ascertain that *murA*(S368P) cells indeed respond to starvation about the same as wt cells.

b) Alternatively, and probably more likely, is that a fraction of (starving) *murA*(S368P) cells lyse, and that the viable cells feed on their released content. In fact, Figure 6—figure supplement 2 indicates that about half of the *murA*(S368P) cells die upon starvation, which is consistent with such a scenario. This will be difficult to detect by OD measurements. So I also recommend that the culture supernatants be tested for cytoplasmic content or, better yet, that starving wt and *murA*(S368P) cultures be tested for continued protein synthesis or for some other marker of metabolic activity, other than PG synthesis.

c) The increased sensitivity of starved *murA*(S368P) cells to the different classes antibiotics (Figure 6) would also be fully compatible with growth of these cells being less sensitive than wt to the starvation conditions used.

[Editors' note: further revisions were requested prior to acceptance, as described below.]

Thank you for resubmitting your work entitled "A cytoplasmic peptidoglycan amidase homologue controls mycobacterial cell wall synthesis" for further consideration at *eLife*. Your revised article has been favorably evaluated by Detlef Weigel as the Senior editor, and three reviewers, one of whom is a member of our Board of Reviewing Editors. The manuscript has been improved but there are some remaining issues, as outlined below.

All three reviewers continue to be excited about the paper, but each agreed that some of the issues initially raised had not been adequately addressed. There were three major concerns and a number of additional concerns. After discussion among the reviewers and editors, it was agreed that all of these concerns would need to be fully addressed if the paper is to be considered further at *eLife*. Particularly for the major issues, additional experiments will be required, not just changes to the language and text. It is not our policy to consider multiple rounds of revision thus without a satisfactory resolution to the major issues below, we regret the paper will not be considered further.

Major issues:

1) The authors did not adequately address original point #7 about showing that MurA is not activated by CwlM(T347A)+PknB *in vitro*. The authors provide some reasons why they didn't do, or didn't include, the experiment, but these were unconvincing. It is a straightforward experiment and must be included. The authors state that they think the result might be that some activation still occurs, but this shouldn't preclude doing and reporting the experiment. In fact, if it turns out that CwlM(T347A) can still activate MurA, the authors will need to significantly revise their model and, consequently, other parts of the paper. For instance, they write when discussing the phenotypes of various mutants that "…it seems likely that absence of phosphorylation on T374 inhibits the essential function of CwlM". And one section states: "We hypothesized that CwlM~P activates MurA and that MurA is less active when CwlM is less phosphorylated, resulting in the inhibition of cell growth seen in the *cwlM* T374A mutant." If the biochemistry indicates that CwlM(T374A) can still activate MurA, such statements and hypotheses would be inaccurate. The authors should also, if the T374A mutant still activates, examine their truncated CwlM construct that lacks multiple putative phosphorylation sites.

2) The original point #8 needs to be more fully addressed. The response addressed the relative stoichiometries of the proteins, but the key issue is the extent of phosphorylation on CwlM. In other words, what percentage of CwlM is phosphorylated in this experiment? As noted in the original point, many bacterial phospho-proteins are not regulated such that 100% of the proteins are phosphorylated, so clarifying what happens to CwlM *in vitro* and *in vivo* is critical. On a related point, in subsection “CwlM~P stimulates the enzymatic activity of MurA” and Figure S5: To establish the MurA/CwlM~P ratio, you need to know the CwlM~P/CwlM ratio first.

3) The section on antibiotic tolerance during starvation (Figure 6) needs additional attention. If we interpret the Materials and methods section correctly, antibiotics were added immediately upon transfer to HdB-C medium. At that point cells are not really starved yet, as indicated by the subsequent ~3-fold increase in OD (Figure 6—figure supplement 1), and they haven't divided-up yet either as indicated by the ~10-fold increase in CFU (Figure 6—figure supplement 1), when antibiotics are absent. The cultures in Figure 6 fail to show an increase in CFU, indicating the antibiotics prevent division of the long pre-shift cells (Figure 6). These cells, in other words, have not had a chance to adjust to carbon-starvation before being hit with antibiotics.

a) This raises the question: Is starvation even relevant here, or would you obtain similar results if you diluted cells in fresh 7H9 medium, and then added the antibiotics?

b) The authors argument about antibiotic tolerance of starved cells would be stronger if cells were first allowed to normally respond to carbon starvation until CFU numbers have stabilized (~20 hrs or so), before being hit with antibiotics.

c) Another advantage is that treated cultures can then be directly compared to untreated control cultures, which are now absent (because they would show a large increase in CFU, I assume). This is relevant in light of the fact that *murA*(S368P) cells show diminished CFU in HdB-C medium, even in the absence of antibiotics (Figure 6—figure supplement 1).

d) Paragraph one, subsection “A regulatory system for PG biosynthesis contributes to drug tolerance”: But you did not really test starved cells. What you tested was antibiotic sensitivity in cells during a transition from nutrient-rich to nutrient-poor conditions, a far more complex situation.

[Editors' note: further revisions were requested prior to acceptance, as described below.]

Thank you for resubmitting your work entitled "A cytoplasmic peptidoglycan amidase homologue controls mycobacterial cell wall synthesis" for further consideration at *eLife*. Your revised article has been favorably evaluated by Detlef Weigel as the Senior editor, a Reviewing editor, and one reviewer.

We continue to be, in principle, inclined to accept the paper, provided you are willing to address the final concern of Reviewer #3, which the Reviewing editor agreed was important for the final version of the paper:

Reviewer #3:

This really is an interesting paper, and the addition of the ClwM phosphoablative mutants in Figure 5 renders the results and model more convincing. I am almost satisfied, except for one item which concerns the new Figure 6.

1) That MurA(S368P) cells fail to control PG synthesis normally is an important point in this paper.

In previous versions this was shown by TADA-labeling of WT and S368P cells that had been starved in HdB-C medium for 4 hrs. Cells were imaged and TADA labeling was quantified. One of the features that made the evidence compelling was the pattern of TADA accumulation in S368P cells (old panel 6C). This showed a clear accumulation of the label at constriction sites, which is where a lot of new PG synthesis takes place, providing visual evidence that most of the TADA-labeling was indeed likely to reflect PG synthesis, as expected.

In the present version of the manuscript, images of TADA-labeled cells are no longer shown. Instead, signals from a fluorescence cell sorter are shown in a new panel (6D), which also now supports the idea that S368P cells over-incorporate TADA during log-phase growth as well as up to 20 hrs after starvation in PBS. This is an important panel but, controls are needed and without cell images as provided in the previous version, virgin readers in particular may not be as convinced that the fluorescence signals plotted in the new panel 6D truly reflect new PG synthesis.

a) Showing some TADA-labeled cells upon starvation in PBS would help convince readers.

b) Paragraph four subsection “MurA activity is downregulated in the transition to starvation”, and Figure 6. The constant fluorescence seen with TADA-treated wt cells could be due to action of Ldts, as authors suggest. It could also be due to auto-fluorescence of the formaldehyde-fixed cells. So, how do curves look when TADA is omitted? It is not obvious this control was done, and it is formally even possible that S368P cells initially just auto-fluoresce more.

c) Also, TADA might have some affinity for some cell feature, without being incorporated in PG. How would curves look when cells are fixed prior to incubation with TADA?

d) Measurements of TADA-, or radiolabeled DAP-, incorporation in SDS-insoluble cell material would be more convincing.

---

## [Author Response]

*Essential revisions:*

1) The claim that CwlM is non-enzymatic is not sufficiently rigorous. It is true that mutations in putative catalytic residues do not cause a loss of function, but this is rather weak; they need to show explicitly that the mutant protein is inactive biochemically. In addition, the mutagenesis is based on the analysis of a truncated B. subtilis protein and the paper cited describes numerous mutations not in the catalytic residues that also inactivated the protein. So, the linkage between specific mutations and lack of catalysis is tenuous. Additionally, according to Deng et al. it does have activity – do you not agree and if so, why?

We have attempted to assay the enzymatic activity of wild-type CwlM without success. However, this negative result in no way proves that the (Deng et al., 2005) study is wrong. We were not able to exactly replicate their methods because the paper omitted many experimental details, so we may not have found the correct conditions yet. However, we feel that the question of whether CwlM is an enzyme is outside the scope of this paper, which is focused on its regulatory function. We rewrote several sections to clarify that we are not disputing that CwlM is an enzyme; we are merely showing that its enzymatic activity is not essential.

While there are other residues that can contribute to catalysis, per (Shida, 2001) the amino acids we mutated are essential. This is further confirmed in another citation (Prigozhin et al., 2013) which used both crystallography and biochemistry to prove that, in a related mycobacterial amidase Rv3717, E200, the functional equivalent of E331 in CwlM, is essential for enzymatic activity.

If the reviewers wish, we can include our negative data that showing a lack of activity for wild-type CwlM in the supplement. We are not including it now because we are not convinced that CwlM is inactive – PG hydrolases are notorious for being difficult to get active *in vitro*, and for often requiring additional factors for activation.

*2) Figure 1: why does depleting CwlM cause cells to die rather than just stop growing? If the model is that CwlM is simply required for promoting cell elongation, then why don't cells lacking CwlM just stop growing? As shown later, when CwlM phosphorylation is downregulated in starvation conditions, cells stop growing, but presumably are not dying, so reconciling these data is important. Figure 1: Do CwlM-depleted cells phenocopy MurA-depleted cells (Xu et al., 2014) or show any shape changes or lysis?*

3) Does the T374A mutant lose viability like the depletion strain? The doubling time is longer (Figure 2) and the cells don't grow polarly (Figure 2) but what about viability?

Our model is that, in the mutant, PG synthesis is no longer tied to the synthesis of other components of the cell wall leading to a loss of cell wall integrity. We have added text to the Discussion to address this issue.

The *cwlM* T374A strain had very low plating efficiency when it was initially constructed for the experiments in Figure 2. We did not measure CFUs in this strain because it is so unstable that we get suppressors within a day of passaging the culture. The low viability of this strain is apparent in the data from Figure 4, where we get almost no transformants that have swapped the wild type allele for the T374A allele in a background with wild type MurA. We were only able to get CwlM T374A allele swaps initially in the strains shown in Figure 2 because we used a parental strain that had the tetR repressor on the L5 vector with CwlM. The *cwlM* promoters both have tetO operators in their promoters. Any transformants that retained the wild-type *cwlM* also retained *tetR*, and the expression of CwlM was repressed. Thus, those strains were built in such a way as to ‘force’ the allele swap and loss of the wild-type allele. In the allele swaps we did for Figure 4, there was no tetR – this allowed us to assess the relative frequency of ‘correct’ allele swaps as a metric for the viability of the resulting strains. We included an explanation of this interpretation in the Discussion, and expanded our explanation of the method Materials and methods.

*4) The authors cite Kall et al. 2007 as evidence that CwlM lacks a secretion signal, but that paper is only a citation to a prediction server and furthermore, running CwlM through that server yields the prediction of a non-cytoplasmic (i.e., secreted) protein. The determination of compartmental localization using the MTSEA/MTSET is also confusing and should be clarified. The authors state in the legend to Figure 2 that "scores >1 are periplasmic, <1 are cytoplasmic", so why does mRFP, unambiguously a cytoplasmic protein, have a value of.84* ± *. 40. In addition to a better explanation of the MTSEA/MTSET approach, the authors should consider a more traditional approach, such as fractionation, unless it is experimentally too demanding. And, perhaps more importantly, is it possible that some CwlM is present in each compartment? Unless this can be categorically ruled out, the authors should adjust their language about the cellular location of CwlM throughout the paper, including in the title.*

The Phobius secretion signal prediction site does not actually predict that CwlM has a secretion signal. It does, for some reason, predict that it is periplasmic, but it predicts that many cytoplasmic proteins are periplasmic, including MurA, GcvH, ParA and FhaA, all of which have established cytoplasmic functions. We were referring to the secretion signal prediction, not the other prediction that Phobius conducts, which seems to be quite faulty. We agree that this is confusing, so we have replaced the Phobius prediction with a reference to the SignalP4.0 site, which also indicates that CwlM lacks a secretion signal.

We have conducted fractionation experiments with CwlM, and we find it in the ‘cytoplasmic’ fraction. However, because there is no way to remove the periplasm as in *E. coli*, that fraction also contains soluble periplasmic proteins that are not tightly associated with the cell wall or the membrane. We were thus not convinced by that experiment, and sought to use other methods. The only published method of determining compartmental location of mycobacterial proteins is PhoA fusions (Lim et al., 1995). Unfortunately, every fusion protein we made to CwlM, including NT, CT and internal fusions of PhoA or fluorescent proteins, were cleaved by the abundant mycobacterial proteases. Thus, we had to develop a new method to determine compartmental localization. We adapted the substituted cysteine accessibility method for this purpose. The cytoplasmic control that we had shown in the original manuscript, mRFP, was partially cleaved, so there were faint extra bands on the gel that confused the result. We have used a new cytoplasmic control, phosphoglycerate mutase (Pgm), which gives us a cleaner result. We also optimized and repeated the assay, and present the new data in Figure 2.

Because there is no perfect method for establishing protein localization, it remains possible that some CwlM is periplasmic, though we suspect that this is unlikely. But our conclusion is that the essential role of CwlM is in the cytoplasm, a conclusion that we feel is consistent with our title.

5) T374 is phosphorylated and the T374A mutant displays some of the phenotypes associated with a cwlM depletion strain. But formally such data alone do not support the conclusion that "CwlM is a cytoplasmic protein whose essential function is activated by phosphorylation." It could be well be that the threonine at position 374 is important for its essential function but not because it must be phosphorylated. At a bare minimum, the authors need to be more precise in their language. Preferably, they would address this experimentally – for example, by testing whether a phosphomimetic (T374D or T374E) allele is hyperactive with respect to activating MurA – has this been tried?

Thank you for the suggestion. Based on this idea, we did construct strains with “phosphomimetic” mutations at the phosphorylated CwlM sites (threonine to aspartates) and we found that the phenotypes were the same as the phospho-ablative mutations. This is not uncommon – amino acid replacements do not always mimic the effect of phosphorylation. We agree with the reviewers that the mutation could alter CwlM function in unexpected ways, and we have changed the text to acknowledge that we cannot be sure of the importance of phosphorylation in regulating CwlM function based on this genetic data alone.

*6) Do constitutively active forms of CwlM rescue the essentiality of pknB mutants? And does the S368P mutation in murA bypass essentiality of pknB?*

This is an interesting question. As described above, we cannot make constitutively active CwlM mutants. But, while the idea that CwlM is the substrate that makes PknB essential is intriguing, we suspect that it is unlikely. PknB has been shown to phosphorylate a broad array of critical proteins and is thought to be a “master regulator” of other protein kinases (Baer et al., 2014). And, while the experiment would be interesting, it is technically difficult and would take a very long time to perform it rigorously. Ultimately, given the low probability of success, we would prefer not to pursue this question. We have rewritten sections in the Discussion to clarify our model that PknB likely regulates many essential proteins.

7) Figure 5: The activation of MurA by CwlM+PknB is convincing, although the authors need to also show that MurA is not activated by CwlM(T347A)+PknB to ensure that PknB isn't having some other effect independent of T347 phosphorylation. Also, does CwlM(T347A) no longer activate MurA, as predicted from the genetics?

We agree that the controls for the biochemistry experiments are critical for interpreting these data. To address this concern we have repeated our controls, moved them from Figure 5—figure supplement 1 to Figure 5, and rewritten several sections to address the controls more clearly. A control with CwlM+PknB+ATP but no MurA shows that CwlM~P alone cannot synthesize EP-UDP-GlcNAc, and a control with PknB+ATP+MurA but no CwlM shows that the activation is specific to CwlM’s phosphorylation, and that PknB does not activate MurA in the absence of CwlM. Finally, the control that was already in Figure 5 is unphosphorylated CwlM that is incubated with PknB but not ATP. Together these controls show that phosphorylated CwlM activates MurA. As the reviewers point out, they do not establish that the activating phosphate is restricted to T374 (*Msmeg*)/ T382 (*Mtb*). We have rewritten sections in the Results and Discussion to make it clear that the other sites on the C-terminal tail are also phosphorylated and that these could contribute to MurA activation.

The reason we did not do the MurA assays with CwlM T382A (the TB analog of CwlMT374A) is that, as the reviewers mention in point 5, the T to A mutation could affect protein function in a way that is independent of phosphorylation. In addition, we did a few experiments that suggest that the other phosphorylation sites we found could still contribute to CwlM’s activation of MurA, though to a lesser extent that T374. We include mention these points in our revised draft:

1) *in vitro* phosphorylated CwlMTB is phosphorylated on the same sites (T382, 384, 386) as in vivo phosphorylated CwlM_SM_ (T374, 376, 378). We performed mass spectrometry to show this, and have added mentions of this in the text and included an annotated alignment in Figure 3—figure supplement 1. This indicates that the pattern of phosphorylation on the C-terminal tail of CwlM is conserved between *Mtb* and *Msmeg*, which along with the strong sequence conservation of the C-terminal tail across actinomyecetes (Figure 2), supports the idea that the entire C-terminal tail is functionally important. Thus we think it likely that the other phosphorylation sites contribute to MurA activation, and we would therefore expect that the CwlM T382A mutant incubated with PknB and ATP would still activate MurA more than the unphosphorylated protein.

2) A strain with alanine substitutions at CwlM T376, T378, K362 and K369 grows more slowly than wild-type. This indicates that phosphorylation at the other sites could, in combination, contribute to the function of CwlM, but that these other sites have less of an effect in isolation. We have included the growth rate data for this strain in Figure 2—figure supplement 1.

Finally, is the basal level activity of the Mtb version of MurA(S368P) significantly higher than that of wt MurA, and/or is unphosphorylated CwlM sufficient to stimulate its activity?

We thank the reviewers for pushing us to do a difficult experiment that really should have been done. While the original biochemistry was performed with Mtb proteins, we were unable to purify MurA S368P from *Mtb* – it was not soluble. In response to this suggestion, we therefore used the *Msmeg* MurA, which is 87% identical and 95% similar to the protein from *Mtb*. We were able to purify soluble *Msmeg* MurA S368P, so we conducted endpoint assays of the WT and mutant MurA in the various CwlM conditions. We present the data in Figure 6—figure supplement 1. The data show that MurA S368P has a higher basal activity level than MurA WT, but that it is also activated by CwlM~P. We have included mention of these data in the text.

8) In the experiment in Figure 5 showing the effect of phosphorylation on the ability of CwlM to activate MurA, they need to characterize the stoichiometry of phosphorylation – the data indicate that this is quite high, but that would be the exception for bacterial phosphoproteins. Also, as noted above, they should try the phosphomimetic mutant of T374 to confirm the genetic data from Figure 2.

We used the TB proteins to do this experiment using endpoint assays, and found, to our surprise, that the stoichiometry of the CwlM~P:MurA reaction appears to be 2:1. We present these data in Figure 5—figure supplement 1 and mention them in the text. This implies that the active complex is possibly a trimer. We hope to explore the nature of this multimer and the detailed molecular mechanism of MurA activation more in future work.

*9) Figure 6 suggests that the change in CwlM phosphorylation is because of a change in PknB levels, but the authors need to verify this by showing that constitutive expression of PknB maintains CwlM phosphorylation upon starvation.*

We have rewritten several sections to clarify our hypothesis that, while PknB is probably primarily responsible for phosphorylating CwlM, there are likely other proteins that dephosphorylate CwlM in starvation. The reduction in CwlM phosphorylation that we observe in Figure 6 is too rapid for transcriptional regulation of pknB alone to be responsible for that change. It is likely that there is a phosphatase that actively removes phosphates from CwlM: this is a question we hope to address in future studies. Overexpressing pknB in starvation may not give the desired result because, in addition to being transcriptionally regulated, pknB is also post-translationally regulated (Baer et al., 2014), so the overexpressed pknB would likely not be active.

*10) Figure 6, subsection “MurA activity is downregulated in starvation” – The TADA incorporation experiments (Figure 6) suggest that the murA(S368P) cells still produce PG upon starvation when the wt cells do not (so much). The authors suggest that growth inhibitory mechanisms are still in effect because cells of both strains become short. Hence, increased TADA incorporation is probably correlated with increased PG turn-over, rather than increased net synthesis, in the murA(S368P) cells. The authors may be correct, but only if cell size is (still) a reliable marker of cell growth.*

*a) It is conceivable that the murA(S368P) mutant cells, though small, do keep on growing and dividing for longer (reach a higher OD) under starvation conditions. For example, depletion of some intermediate of LipidII synthesis might normally act as a general signal to stop/slow growth in wt cells, and this might no longer be very effective in the mutant. I recommend that the authors do some simple growth curves to acertain that murA(S368P) cells indeed respond to starvation about the same as wt cells.*

We agree with the reviewers that this seems very likely. We have done these growth curves in carbon-limited media and see a slight but statistically insignificant difference, which we have now added to Figure 6—figure supplement 1. The *murA* S368P strain may reach a slightly higher OD during the transition to starvation, before starting to die more in stationary phase. We have mentioned this in the Results and the Discussion.

*b) Alternatively, and probably more likely, is that a fraction of (starving) murA(S368P) cells lyse, and that the viable cells feed on their released content. In fact, Figure 6—figure supplement 1 indicates that about half of the murA(S368P) cells die upon starvation, which is consistent with such a scenario. This will be difficult to detect by OD measurements. So I also recommend that the culture supernatants be tested for cytoplasmic content or, better yet, that starving wt and murA(S368P) cultures be tested for continued protein synthesis or for some other marker of metabolic activity, other than PG synthesis.*

We performed this experiment by blotting the *murA* S368P and WT culture supernatants for rpoB during carbon starvation. We found that the mutant culture had more rpoB in the supernatant than the WT, and have reported this in the text and in Figure 6—figure supplement 1. We also include an acknowledgment that the higher TADA staining could be due to the extra nutrients available to the mutant cells due to lysis.

*c) The increased sensitivity of starved murA(S368P) cells to the different classes antibiotics (Figure 6) would also be fully compatible with growth of these cells being less sensitive than wt to the starvation conditions used.*

We agree that this is a possibility, and have added text about this in the Discussion.

[Editors' note: further revisions were requested prior to acceptance, as described below.]

*All three reviewers continue to be excited about the paper, but each agreed that some of the issues initially raised had not been adequately addressed. There were three major concerns and a number of additional concerns. After discussion among the reviewers and editors, it was agreed that all of these concerns would need to be fully addressed if the paper is to be considered further at eLife. Particularly for the major issues, additional experiments will be required, not just changes to the language and text. It is not our policy to consider multiple rounds of revision thus without a satisfactory resolution to the major issues below, we regret the paper will not be considered further.*

*Major issues:*

*1) The authors did not adequately address original point #7 about showing that MurA is not activated by CwlM(T347A)+PknB* in vitro. *The authors provide some reasons why they didn't do, or didn't include, the experiment, but these were unconvincing. It is a straightforward experiment and must be included. The authors state that they think the result might be that some activation still occurs, but this shouldn't preclude doing and reporting the experiment. In fact, if it turns out that CwlM(T347A) can still activate MurA, the authors will need to significantly revise their model and, consequently, other parts of the paper. For instance, they write when discussing the phenotypes of various mutants that "…it seems likely that absence of phosphorylation on T374 inhibits the essential function of CwlM". And one section states: "We hypothesized that CwlM~P activates MurA and that MurA is less active when CwlM is less phosphorylated, resulting in the inhibition of cell growth seen in the cwlM T374A mutant." If the biochemistry indicates that CwlM(T374A) can still activate MurA, such statements and hypotheses would be inaccurate. The authors should also, if the T374A mutant still activates, examine their truncated CwlM construct that lacks multiple putative phosphorylation sites*.

We have now conducted MurA activation assays with CwlM T382A (The TB analog of CwlM T374A from Msmeg) and with CwlM T382A T384A T386A. The results with the *Mtb* proteins mirror the results from the *Msmeg* genetic experiments: CwlM T382A in an active kinase reaction has very low MurA stimulation activity, and CwlM T382A T384A T386A behaves just like unphosphorylated CwlM. We include these data in Figure 5.

*2) The original point #8 needs to be more fully addressed. The response addressed the relative stoichiometries of the proteins, but the key issue is the extent of phosphorylation on CwlM. In other words, what percentage of CwlM is phosphorylated in this experiment? As noted in the original point, many bacterial phospho-proteins are not regulated such that 100% of the proteins are phosphorylated, so clarifying what happens to CwlM* in vitro *and* in vivo *is critical. On a related point, in subsection “CwlM~P stimulates the enzymatic activity of MurA” and Figure 5—figure supplement 1: To establish the MurA/CwlM~P ratio, you need to know the CwlM~P/CwlM ratio first.*

We used a 2D gel to assess how much his-CwlM is phosphorylated in the in vitro assays. Upon phosphorylation by PknB, we see a shift of the entire his-CwlM spot to a lower isoelectric point. We interpret this to mean that the vast majority of the protein is phosphorylated on at least one site. Because the 2D gel cannot tell us exactly how much his-CwlM is in each of the 7 possible combinatorial phosphorylation configurations, there remain uncertainties. We appreciate that these uncertainties mean that the kinetic constants we present in Figure 5 actually represent a mixture of different rates: it is likely that CwlM phosphorylated at T382 has a different rate than CwlM phosphorylated at T384 and T386. We include language to explain this, and to emphasize that the constants are presented for broad comparisons, and are not intended to represent the precise rate of a single intermolecular interaction.

*3) The section on antibiotic tolerance during starvation (Figure 6) needs additional attention. If we interpret the Materials and methods section correctly, antibiotics were added immediately upon transfer to HdB-C medium. At that point cells are not really starved yet, as indicated by the subsequent ~3-fold increase in OD (Figure 6—figure supplement 1), and they haven't divided-up yet either as indicated by the ~10-fold increase in CFU (Figure 6—figure supplement 1), when antibiotics are absent. The cultures in Figure 6 fail to show an increase in CFU, indicating the antibiotics prevent division of the long pre-shift cells (Figure 6). These cells, in other words, have not had a chance to adjust to carbon-starvation before being hit with antibiotics.*

*a) This raises the question: Is starvation even relevant here, or would you obtain similar results if you diluted cells in fresh 7H9 medium, and then added the antibiotics?*

*b) The authors argument about antibiotic tolerance of starved cells would be stronger if cells were first allowed to normally respond to carbon starvation until CFU numbers have stabilized (~20 hrs or so), before being hit with antibiotics.*

*c) Another advantage is that treated cultures can then be directly compared to untreated control cultures, which are now absent (because they would show a large increase in CFU, I assume). This is relevant in light of the fact that murA(S368P) cells show diminished CFU in HdB-C medium, even in the absence of antibiotics (Figure 6—figure supplement 1).*

*d) Paragraph one, subsection “A regulatory system for PG biosynthesis contributes to drug tolerance”: But you did not really test starved cells. What you tested was antibiotic sensitivity in cells during a transition from nutrient-rich to nutrient-poor conditions, a far more complex situation.*

The reviewers’ comments made us realize that there were several confusing things about Figure 6. First, we had done starvation experiments in both PBS starvation and in carbon-only starvation in HdB media. The no-drug CFU experiment was also done in low carbon media, while the drug experiments were done in no carbon media. We decided to redo all the experiments in PBS starvation for the sake of consistency.

Secondly, we needed to identify the conditions in which the post-translational regulation of MurA affects cell physiology. We have now used the D-alanine dye TADA to assess PG metabolism in the WT and the over-active *murA* S368P strain in log phase and during 24 hours of PBS starvation. We used flow cytometry to measure the level of fluorescent staining with this dye at several time points (Figure 6). We found that the difference in TADA staining is highest in log phase and early starvation, and decreases in late starvation. We think this means that the post-translational regulation of MurA is likely to be important during environmental transitions.

Finally, we redid the CFU killing experiments in log phase growth in 7H9, in the transition to PBS starvation, and in deep starvation. The results are what we expect from the TADA staining experiment: we see the biggest defect in viability of the *murA* S368P strain in log phase and during the transition to starvation. In the 7H9 experiments the data are difficult to interpret because some of the replicates developed antibiotic resistance over the course of the experiment. We changed the wording throughout to indicate that the MurA regulatory system is important during the transition to starvation, and we see differences in antibiotic tolerance then.

When we did the starvation experiments with carbon starvation in HdB we had seen a viability/ growth defect of the *murA* S368P mutant during starvation alone. However, we did not observe this defect reproducibly in PBS starvation, so we have removed mention of this defect throughout, and we focus only on the antibiotic sensitivity phenotype.

[Editors' note: further revisions were requested prior to acceptance, as described below.]

*This really is an interesting paper, and the addition of the ClwM phosphoablative mutants in Figure 5 renders the results and model more convincing. I am almost satisfied, except for one item which concerns the new Figure 6.*

*1) That MurA(S368P) cells fail to control PG synthesis normally is an important point in this paper.*

*In previous version, this was shown by TADA-labeling of WT and S368P cells that had been starved in HdB-C medium for 4 hr. Cells were imaged and TADA labeling was quantified. One of the features that made the evidence compelling was the pattern of TADA accumulation in S368P cells (old panel 6C). This showed a clear accumulation of the label at constriction sites, which is where a lot of new PG synthesis takes place, providing visual evidence that most of the TADA-labeling was indeed likely to reflect PG synthesis, as expected.*

*In the present version of the manuscript, images of TADA-labeled cells are no longer shown. Instead, signals from a fluorescence cell sorter are shown in a new panel (6D), which also now supports the idea that S368P cells over-incorporate TADA during log-phase growth as well as up to 20 hrs after starvation in PBS. This is an important panel but, controls are needed and without cell images as provided in the previous version, virgin readers in particular may not be as convinced that the fluorescence signals plotted in the new panel 6D truly reflect new PG synthesis.*

*a) Showing some TADA-labeled cells upon starvation in PBS would help convince readers.*

Figure 6.

*b) Paragraph four subsection “MurA activity is downregulated in the transition to starvation”, and Figure 6. The constant fluorescence seen with TADA-treated wt cells could be due to action of Ldts, as authors suggest. It could also be due to auto-fluorescence of the formaldehyde-fixed cells. So, how do curves look when TADA is omitted? It is not obvious this control was done, and it is formally even possible that S368P cells initially just auto-fluoresce more.*

Figure 6—figure supplement 1.

*c) Also, TADA might have some affinity for some cell feature, without being incorporated in PG. How would curves look when cells are fixed prior to incubation with TADA?*

Figure 6—figure supplement 1.

*d) Measurements of TADA-, or radiolabeled DAP-, incorporation in SDS-insoluble cell material would be more convincing.*

Figure 6—figure supplement 1. You can see from the inset that, in the mutant, the SDS insoluble fraction stains more brightly even by eye.